# Ultrafast upconversion superfluorescence with a sub-2.5 ns lifetime at room temperature

Mengwei Zhou[1,2,3], Ping Huang [1,2,3] ✉, Xiaoying Shang [1,2] ✉, Ruihuan Zhang[1], Wen Zhang[1,3], Zhiqing Shao[1,3], Shuo Zhang[1], Wei Zheng [1,2,3] ✉ & Xueyuan Chen [1,2,3] ✉

Photon upconversion through lanthanide-doped nanoparticles is of great significance for various applications. However, the current development of upconversion nanoparticles is hindered by the low quantum efficiency and long radiative lifetimes of lanthanide ions, restricting their applications in time-dependent nanophotonics. Herein, we report ultrafast upconversion superfluorescence with a lifetime of sub-2.5 ns in lanthanide-doped nanoparticles at room temperature. Upon excitation with an 800-nm fs-pulsed laser, we achieve a large number ($N = 912$) of correlated dipoles in $Nd^{3+}$-concentrated nanoparticles, resulting in collective coherent emission with two orders of magnitude amplification in intensity and more than three orders of magnitude improvement in the radiative decay rate. Furthermore, we demonstrate that the control of excitation power and emitting sample length enables the lifetime manipulation of upconversion emission in a wide range from μs to sub-ns, accompanied by the typical superfluorescence signature of Burnham-Chiao ringing. These findings may benefit applications in many advanced technologies such as quantum counting and high-speed super-resolution bioimaging.

Lanthanide ($Ln^{3+}$)-doped upconversion nanoparticles (UCNPs), capable of converting low-energy near-infrared irradiation into high-energy ultraviolet and visible emissions, have attracted considerable attention for a variety of optoelectronic and biomedical applications owing to their outstanding optical properties such as large anti-Stokes-like shift, sharp emission peaks, and high photochemical stability[1–5]. Despite the great prospect, the current development of $Ln^{3+}$-doped UCNPs is hindered by the low upconversion (UC) efficiency, small absorption cross-section, and slow radiative decay rate resulting from the parity-forbidden 4 f → 4 f transition of $Ln^{3+}$ [6–9]. Specifically, the long radiative decay time of UCNPs on the microsecond to millisecond scale has severely restricted their applications in many advanced

technologies such as high-speed super-resolution bioimaging and ultrafast signal transduction in which a fast radiative decay rate is essential[10–14]. To circumvent these limitations, plasmonic nanocavities can be coupled with UCNPs to boost the spontaneous emission (SE) rate and enhance the absorption and emission efficiencies of $Ln^{3+}$, resulting in a UC superburst with a lifetime of down to 50 ns[15,16]. However, the implantation of plasmonic nanocavities is complicated and requires meticulous design with increased dimensions of the emitting system, which is not favorable for specific biomedical and nanophotonic applications.

Superfluorescence (SF) is a quantum optical phenomenon in which an ensemble of emitters is coherently coupled to generate a

[1]State Key Laboratory of Structural Chemistry, Fujian Key Laboratory of Nanomaterials, and CAS Key Laboratory of Design and Assembly of Functional Nanostructures, Fujian Institute of Research on the Structure of Matter, Chinese Academy of Sciences, Fuzhou, China. [2]Fujian Science & Technology Innovation Laboratory for Optoelectronic Information of China, Fuzhou, China. [3]University of Chinese Academy of Sciences, Beijing, China. ✉e-mail: huangping09@fjirsm.ac.cn; shangxiaoying@fjirsm.ac.cn; zhengwei@fjirsm.ac.cn; xchen@fjirsm.ac.cn

short but intense burst of light[17–19]. The hallmark of SF is the collective, synergistic photon emission from the photo-excited aligned dipoles with integrated strengths, therefore it provides significantly stronger emission with a greatly improved radiative decay rate as compared to that of SE based on individual uncorrelated dipoles[20]. This feature makes SF highly attractive in quantum optics and ultrafast photonics[21]. Nevertheless, because of the environmental perturbation and ultra-short dephasing time of coherence, the realization of SF is challenging and has been confined to a few atomic gases and solid-state matrices such as superlattices of perovskite nanocrystals and stacked semi-conductor quantum wells with long-range orderliness at cryogenic temperatures and under high magnetic fields[22–24]. Recently, Lim and Han et al. discovered the room-temperature (RT) anti-Stokes-like UC-SF with a lifetime of 46 ns in Nd[3+]-enriched UCNPs, which overcame the deficiency of normal UC luminescence (UCL) with respect to its long lifetime (μs–ms)[25]. In comparison with the existing downshifting SF medium that uses the entire nanoparticle as an emitter, each Ln[3+] ion in a single UCNP can function as an individual emitter, which interacts with each other to establish coherence without the need of sophisti-cated design like superlattice and resonant-cavity enhancement. However, the number of coherently coupled dipoles ($N$) in the repor-ted Nd[3+]-enriched UCNPs was only 11, which is not ideal for the col-lective SF, because the SF emission intensity $I \propto N^2$ and the radiative decay time of SF $\tau_{SF} \propto \tau_{SE}/N$ (where $\tau_{SE}$ is the spontaneous decay time)[26]. Moreover, the fundamental photophysics of UC-SF including the UC-SF dynamics, the excitation power dependence, and the emitting sample length dependence remains largely unexplored.

Herein, we report ultrafast UC-SF in Nd[3+]-concentrated NaNdF$_4$: $x$% Nd[3+]@NaYF$_4$ core-shell UCNPs upon excitation with an 800-nm fs-pulsed laser at RT and free space without the use of plasmonic nano-cavity. Owing to the strong coupling of Nd[3+] under a high radiation field provided by the fs-pulsed laser excitation, a number of coherently coupled dipoles of up to 912 was realized, resulting in three orders of magnitude improvement in the radiative decay rate of Nd[3+] as com-pared to that of normal UCL, in parallel with a record-short lifetime of sub-2.5 ns that had never been achieved before. The effects of the excitation power, Nd[3+] concentration, and emitting sample length on UC-SF as well as its kinetics were investigated in detail. All the sig-natures of SF including the power-dependent build-up and decay times, fourth-order power dependence of the two-photon UC emis-sion, and UC-SF oscillation were observed, providing solid evidence for UC-SF in the Nd[3+]-concentrated system. The breakdown of radiative lifetime of Ln[3+] from the μs–ms scale to sub-ns through UC-SF paves the straightforward way for Ln[3+] luminescence in ultrafast optics toward various state-of-the-art applications.

## Results

Figure 1a illustrates the build-up process of SF in an ensemble of $N$ emitters concentrated in a region with a volume V smaller than $\lambda^3$, where $\lambda$ is the emission wavelength of the emitter. Upon excitation with an intense pulsed laser, a total population inversion is established between the ground state and excited state of the emitters, generating a group of uncorrelated dipoles with randomly distributed phases in the system. The uncorrelated dipoles trigger the SE. Once the SE starts, the quantum fluctuations of the electromagnetic field of the vacuum act on each independent emitter, leading to the coherent coupling of the photo-excited dipoles through spontaneous synchronization[27]. As a result, the initially uncorrelated dipoles become highly correlated and are aligned in phase, appearing like a macroscopic giant dipole. The resulting coherent quantum state has several orders of magnitude larger dipole strength than the incoherent state, giving rise to a burst of luminescence (namely, SF) with extremely high intensity and ultrafast decay time distinguishing from those of SE. The conditions for realizing SF are very stringent, requiring high-power pulsed

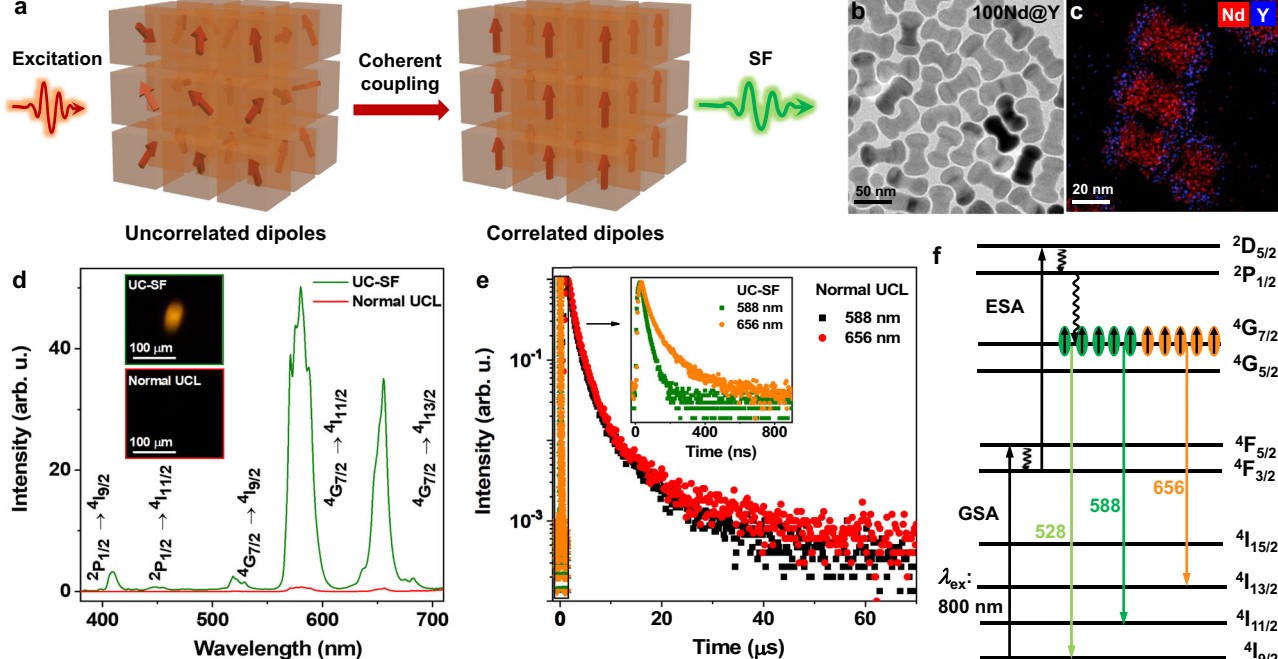

**Fig. 1 | UC-SF of Nd[3+] in NaNdF$_4$@NaYF$_4$ (100Nd@Y) UCNPs. a** Schematic of the build-up process of SF. The initially uncorrelated photo-excited dipoles with ran-domly distributed phases become correlated in phase through coherent coupling, forming a macroscopic giant dipole to generate SF. **b** TEM image and **(c)** EDX elemental mapping of 100Nd@Y UCNPs. **d** Normal UCL and UC-SF spectra of 100Nd@Y UCNPs upon 808-nm CW and 800-nm fs-pulsed laser excitation with an average power density of ~1.10 kW cm[−2], respectively. The insets show the microscopic images for UC-SF and normal UCL of the UCNP assemblies. **e** Normal UCL and UC-SF decay curves of 100Nd@Y UCNPs by monitoring the $^4G_{7/2} \to \ ^4I_{11/2}$ and $^4G_{7/2} \to \ ^4I_{13/2}$ emissions of Nd[3+] at 588 and 656 nm, respectively. The inset shows the enlarged UC-SF decay curves of Nd[3+] at 588 and 656 nm. **f** Energy levels and electronic transitions of Nd[3+] for normal UCL and UC-SF in 100Nd@Y UCNPs. GSA and ESA denote the ground-state absorption and excited-state absorption, respectively.

excitation to create population inversion for efficient dipole-dipole phase locking, a high density of emitters in a small volume to guarantee their collective interaction with the radiation field, small inhomogeneous emission-line broadening, and a long dephasing time of the emitters to safeguard the coherent states against thermal and environmental perturbation[28].

To fulfill the requirements for achieving UC-SF, we synthesized $Nd^{3+}$-concentrated $NaNdF_4$ UCNPs and coated them with a layer of inert $NaYF_4$ shell to inhibit the surface quenching effect on $Nd^{3+}$ luminescence. Structural and morphology characterizations through XRD, TEM, and energy-dispersive X-ray (EDX) elemental mapping revealed that the as-synthesized $NaNdF_4@NaYF_4$ (100Nd@Y) core-shell UCNPs had a hexagonal $Na(Nd/Y)F_4$ phase and a dumbbell-like morphology with $NaNdF_4$ nanorods sandwiched by $NaYF_4$ nanoplates (Fig. 1b, c and Supplementary Figs. 1, 2)[29]. The average length and diameter of 100Nd@Y UCNPs were calculated to be ~54.9 and ~30.7 nm, respectively (Supplementary Fig. 3). All the dimensions of the UCNPs are smaller than the emission wavelengths of $Nd^{3+}$ in the visible region, satisfying the condition of small-sample SF system $(V < \lambda^3)$[27,30].

For UC-SF measurements, a customized microscopic spectroscopy system was built and equipped with both continuous-wave (CW) and fs-pulsed laser as two independent excitation sources (Supplementary Fig. 4). We first measured the normal UCL spectrum of 100Nd@Y UCNPs under CW laser excitation at 808 nm with a power density of ~1.10 kW cm⁻². As shown in Fig. 1d and Supplementary Fig. 5, the UCNPs exhibited weak luminescence with characteristic emission peaks at 528, 588, and 656 nm, corresponding to the electronic transitions of $Nd^{3+}$ from $^4G_{7/2}$ to $^4I_{9/2}$, $^4I_{11/2}$, and $^4I_{13/2}$, respectively. The inefficient UCL of $Nd^{3+}$ under CW laser excitation is not unexpected since $Nd^{3+}$ is not a typical UC emitter due to the dense energy levels of $Nd^{3+}$ that impose deleterious nonradiative energy losses through cross-relaxation and energy migration among $Nd^{3+}$ to the surface and lattice defects[31]. By contrast, upon 800-nm fs-pulsed laser excitation with an equivalent power density at average (~1.10 kW cm⁻²), the UCNPs displayed bright luminescence (insets of Fig. 1d), with 70 times enhancement in UCL intensity and the emergence of new emission peaks from high energy level of $Nd^{3+}$ at 409 nm ($^2P_{1/2} \rightarrow {}^4I_{9/2}$) and 449 nm ($^2P_{1/2} \rightarrow {}^4I_{11/2}$). Specifically, we found that the decay times of the $^4G_{7/2} \rightarrow {}^4I_{11/2}$ (588 nm) and $^4G_{7/2} \rightarrow {}^4I_{13/2}$ (656 nm) transitions of $Nd^{3+}$ under fs-pulsed laser excitation were different and abnormally shortened to 10.7 and 24.8 ns, respectively (inset of Fig. 1e). These observations are in stark contrast to the normal UCL lifetimes of $Nd^{3+}$ under ns-pulsed laser excitation, where $Nd^{3+}$ displayed an identical UCL decay time ($\tau_{SE}$) of 2.28 μs at 588 and 656 nm (Fig. 1e). Generally, the decay times of the parity-forbidden $4f \rightarrow 4f$ transitions of $Ln^{3+}$ ions are on the μs–ms range and the decay times of the UCL from the same emitting level should be identical, because each transition dipole shares the same deexcitation channels of the emitting level[32]. Additionally, in normal UCL, the enhancement of UCL intensity is usually accompanied by the lengthening of UCL lifetime due to the suppressed nonradiative relaxation, while the shortening of UCL lifetime indicates accelerated nonradiative relaxation of excited $Ln^{3+}$ ions through energy transfer to the lattice or surface defects, which results in decreased UCL intensity[33,34]. Hence, the observation of remarkably enhanced UCL intensity along with decay times on the ns scale may result from the burst of radiative transition rate of $Nd^{3+}$ instead of the suppressed nonradiative relaxation, signifying that the upconverted emission of 100Nd@Y under fs-pulsed laser excitation is not a normal UCL process. Instead, it behaves like UC-SF, namely, collective emission of coherent dipoles, wherein the decay time of the emitter $\tau_{SF} \propto \tau_{SE}/N$ (where $N$ is the number of emitters coupled in the coherent state). Actually, $Ln^{3+}$-doped UCNPs are a multilevel SF system because of abundant electronic transitions within a single $Ln^{3+}$ ion. For SF in a multilevel system, the emission can occur successively on two

cascading transitions at two different frequencies or two transitions with different frequencies sharing a common upper level, which can be in competition for the depletion of this level, due to the different frequencies or different polarizations of the radiation fields[30]. The transitions from $^4G_{7/2}$ to $^4I_{11/2}$ and $^4I_{13/2}$ of $Nd^{3+}$ are the Λ-type competing transitions, in which the excited state can decay to multiple ground states with different frequencies[35]. Therefore, the distinct decay dynamics of the $^4G_{7/2} \rightarrow {}^4I_{11/2}$ and $^4G_{7/2} \rightarrow {}^4I_{13/2}$ transitions of $Nd^{3+}$ indicates the presence of two independent ensembles of coherently coupled dipoles, which may compete for the depletion of the shared upper level of $^4G_{7/2}$ (Fig. 1f). This phenomenon further confirms the establishment of the dipole-dipole correlations for UC-SF in 100Nd@Y UCNPs upon fs-pulsed laser excitation.

To gain deep insights into the UC-SF dynamics of $Nd^{3+}$, we recorded the excitation power-dependent transient kinetic traces of 100Nd@Y UCNPs. As shown in Fig. 2a, the UC-SF intensity increased gradually with increasing the excitation power density from 0.71 to 2.09 kW cm⁻², concurrent with a remarkable shortening in the decay time of $Nd^{3+}$ at 588 nm from 37.4 ns to 2.5 ns (Fig. 2b and Supplementary Fig. 6), due to the increased number of coupled dipoles in the coherent state. According to the equation $\tau_{SF} \propto \tau_{SE}/N$, the number of coherently coupled dipoles for the $^4G_{7/2} \rightarrow {}^4I_{11/2}$ transition of $Nd^{3+}$ was estimated to be 61, 67, 174, 326, and 912 at the excitation power density of 0.71, 0.75, 1.07, 1.38, and 2.09 kW cm⁻², respectively. Such a large number ($N = 912$) of coherently coupled dipoles achieved in 100Nd@Y UCNPs is ~83 times larger than that ($N = 11$) reported by Lim and Han et al. in $NaYF_4:Yb,Er@NaYF_4:Yb@NaNdF_4:Yb$ core-shell-shell UCNPs under ns-pulsed laser excitation[25]. The ultrashort UC-SF decay time of 2.5 ns is also ~18 times shorter than that (46 ns) of Lim and Han's report. We speculate that the improved UC-SF properties achieved in our 100Nd@Y UCNPs lie in the use of small-sample SF system along with fs-pulsed laser excitation, which resulted in significantly enhanced radiation field and consequently increased number of coherently coupled dipoles as compared to those in larger samples with ns-pulsed laser excitation (Supplementary Figs. 7, 8). Notably, the UC-SF decay time of 100Nd@Y UCNPs can be further reduced to below 2.5 ns by increasing the $N$ value upon increasing the excitation power density, which was restricted by the detection limit of our instrument (Supplementary Fig. 9). In addition to the excitation power-dependent decay time, another signature of UC-SF regarding the power dependence of the coherent state build-up time is expected because of the time required to achieve phase synchronization among the initially incoherent dipoles. As shown in Fig. 2c, d, the UC-SF build-up (or delay) time ($\tau_D$) of $Nd^{3+}$ decreased gradually from 25.9 ns to 16.5 ns with an increase in the excitation power density from 0.71 to 2.09 kW cm⁻² (Supplementary Fig. 10). This observation is consistent with previous findings in perovskite nanocrystal films and superlattices[36,37], wherein the delay time of the downshifting SF of the excitons underwent a decrease with increasing the excitation power density due to the improved coherent state upon high-power excitation.

The UC-SF dynamics of 100Nd@Y UCNPs can be described based on the Dicke model considering an ensemble of $N$ two-level identical atoms (emitters) (Fig. 2e)[27]. The upper and lower states of each emitter are represented by $|e\rangle$ and $|g\rangle$. At time $t = 0$, all the emitters are initially excited with the angular momentum of the system $J = N/2$, assuming each two-level atom has a fictitious spin of 1/2. The radiative transition rate of the $N$-atom system can be written as:

$$W_N = \Gamma(J + M)(J - M + 1) \tag{1}$$

where $\Gamma$ is the atomic natural linewidth corresponding to the transition from $|e\rangle$ to $|g\rangle$. At initial time ($t = 0$), $J = M = N/2$ and the radiative transition rate $W_N = \Gamma N$, which means that the emission of the $N$-atom system starts from SE. With the time evolution, the system appears as a cascade emission down through a "ladder" of $2J + 1$ equidistant levels,

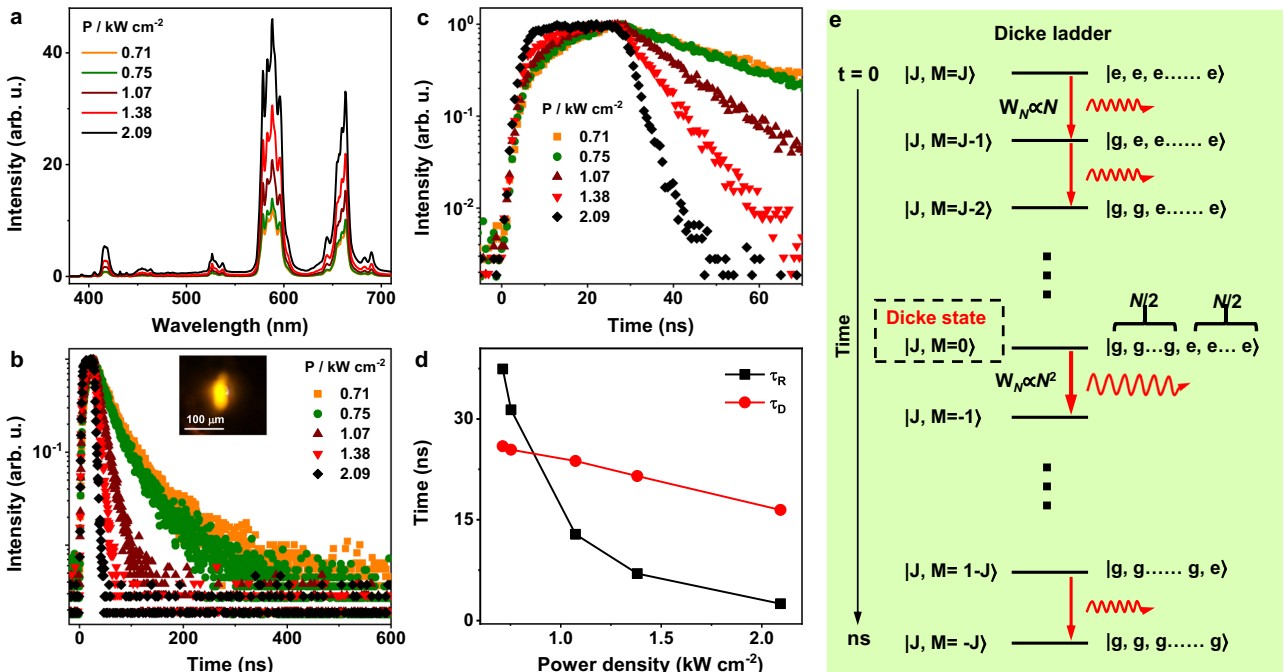

**Fig. 2 | Power-dependent UC-SF spectra and decay curves of 100Nd@Y UCNPs.**
**a** UC-SF spectra and (**b**) decay curves ($\lambda_{em}$ = 588 nm) of 100Nd@Y UCNPs upon fs-pulsed laser excitation at 800 nm with different power densities. The inset in (**b**) shows the UC-SF image of the UCNP assemblies. **c** Power-dependent UC-SF decay curves ($\lambda_{em}$ = 588 nm) of 100Nd@Y UCNPs at the initial stage, showing the decreased delay time with increasing the excitation power density. **d** The delay time ($\tau_D$) and radiative decay time ($\tau_R$) of $Nd^{3+}$ at 588 nm as a function of excitation power density. **e** Schematic of the SF dynamics of the collective $N$-atom SF system based on the Dicke model.

with $W_N$ increasing from $\Gamma N$ to the maximum value $\Gamma(N^2/4)$ at the half-deexcited state when M = 0. The half-deexcited state $|N/2, 0\rangle$ is defined as Dicke superradiant state[27], wherein the macroscopically coherent transition dipole is established. These processes are very fast and can be reflected as a delay time in the transient kinetic trace of SF. Typically, the delay time ($\tau_D$) of the ideal small-sample SF is longer than its radiative decay time ($\tau_R$) with the estimation of $\tau_D/\tau_R \propto \log_{10}(N)$[26], suggesting that the UC-SF of 100Nd@Y UCNPs with $\tau_D$ = 16.5 ns and $\tau_R$ = 2.5 ns excited at 2.09 kW cm$^{-2}$ is a pure SF process.

According to the theory of SF, the short distance ($r_{ij}$) among emitters is crucial for the efficient dipole-dipole correlation, while the coherent dipole dephasing time is inversely proportional to $r_{ij}^3$[30]. Therefore, the concentration of the emitters has a significant impact on the SF properties. To investigate the effect of $Nd^{3+}$ concentration on the UC-SF properties, we synthesized NaYF$_4$: $x$ mol%$Nd^{3+}$@NaYF$_4$ ($x$Nd@Y) ($x$ = 2, 25, 50, 75, and 100) core-shell UCNPs with different $Nd^{3+}$ concentrations. XRD, TEM, and EDX elemental mappings confirmed the hexagonal phase and core-shell structure of the UCNPs with dimensions in the range of 46.5–173 nm and morphology evolving from nano-dumbbells to nanoplates as the $Nd^{3+}$ concentration decreased from 100 mol% to 2 mol% (Fig. 3a–d and Supplementary Figs. 11–14), fulfilling the condition of small-volume UC-SF system. All the UCNPs exhibited significantly enhanced UCL intensity upon fs-pulsed laser excitation relative to that upon CW laser excitation with an equivalent power density at average, except for UCNPs with 2 mol% $Nd^{3+}$ (Supplementary Fig. 15). The UCL enhancement factors were calculated to be 1, 44, 94, 86, and 70 for UCNPs with $Nd^{3+}$ concentration of 2, 25, 50, 75, and 100 mol%, respectively (Fig. 3e). The negligible UCL enhancement observed in 2Nd@Y UCNPs is attributed to the large interionic distance between $Nd^{3+}$ ions, which is not beneficial to the coherent coupling of the transition dipoles (Fig. 3f). It is worth mentioning that higher UCL enhancement can be achieved by fine-tuning the excitation wavelength of the fs-pulsed laser since 800 nm is not the

best for $Nd^{3+}$ excitation. These results reveal that 50Nd@Y UCNPs with 50 mol% of $Nd^{3+}$ in the core exhibited the most efficient UC-SF, in which the formation and dephasing time of the coherently coupled dipoles was well balanced. As a result, the excitation power density required for building up the efficient macroscopic giant dipole can be reduced from 2.09 kW cm$^{-2}$ in 100Nd@Y UCNPs to 1.53 kW cm$^{-2}$ in 50Nd@Y UCNPs, whereby ultrafast UC-SF with a sub-2.5 ns decay time approaching the detection limit of our instrument was realized (Fig. 3g). Furthermore, we measured the power-dependent UC-SF spectra of 50Nd@Y UCNPs (Supplementary Fig. 16). The UCNPs demonstrated the power dependence with a slope of 3.44 and 3.28 at the low power region (<1.34 kW cm$^{-2}$) for the $Nd^{3+}$ emissions at 588 nm and 656 nm, respectively, whereas the slopes decreased to 1.53 and 1.52 when the excitation power density exceeded 2.09 kW cm$^{-2}$ due to the UC saturation effect (Fig. 3h). Our measurements are in good agreement with the expected power dependence for two-photon UC-SF with $I_{UC-SF} \propto N^2 \propto P^4$ (where $P$ is the excitation power density)[25], distinguishing from that of normal UCL with $I_{UCL} \propto P^2$ (Fig. 3i).

As a characteristic of SF, oscillatory SF termed as Burnham-Chiao ringing is expected in the decay curve at the condition of a large active volume ($V > \lambda^3$)[38,39]. Burnham-Chiao ringing is based on the reabsorption/emission of the sample and reflects the coherent Rabi-type interaction between the propagating SF pulse and the sample[36]. According to Arecchi-Courtens limit[38], there is a maximum cooperation number $N_c$ in the large sample case. The number $N_c$ determines the emitting sample length, beyond which specific propagation effect plays a major role[40]. To obtain the oscillation information and validate the UC-SF of $Nd^{3+}$, we increased the emitting sample length ($L_{em}$) of the system by modulating the diameter of the excitation spot. The emission at the excitation spot diameter of 56.3 µm displayed a single-pulse decay with a fitted lifetime of 2.6 ns (Fig. 4a), indicative of pure UC-SF of the system. As the excitation spot increased to 105.7 and 163.0 µm, the effective emitting sample length increased accordingly, and the

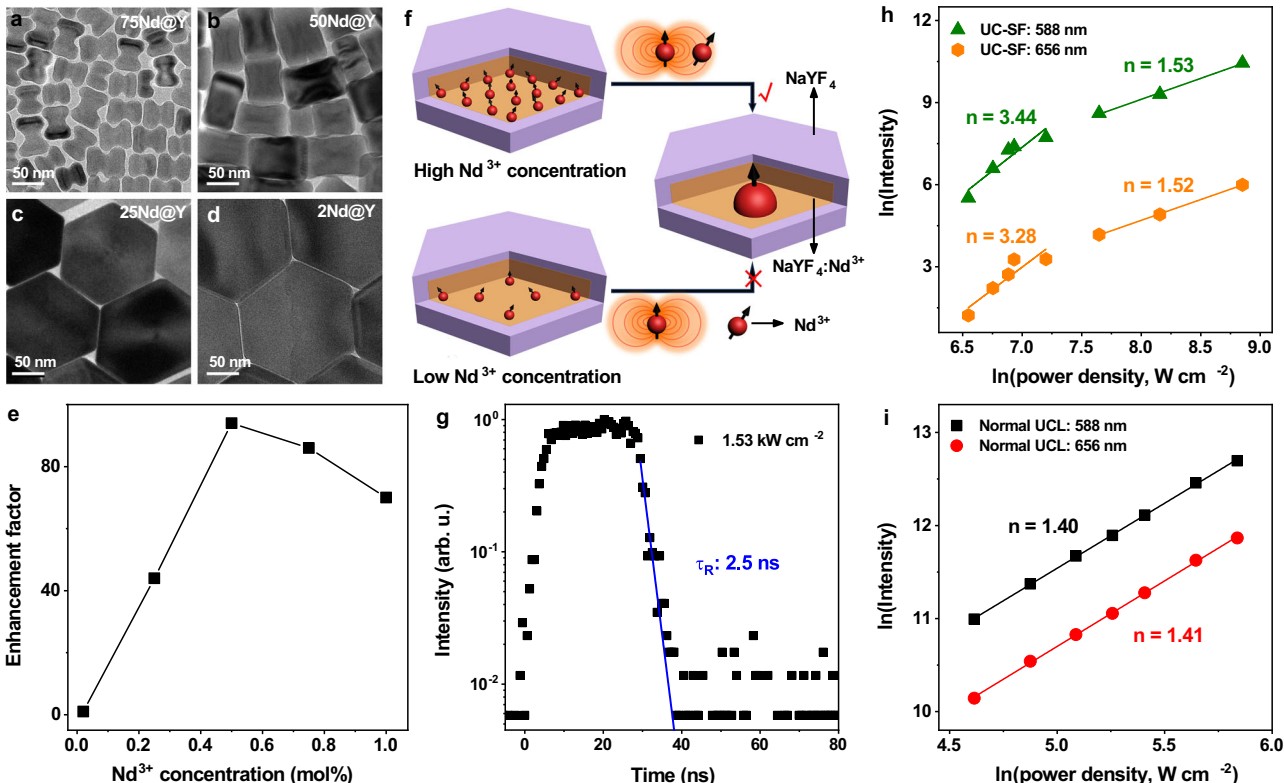

**Fig. 3 | Nd³⁺ concentration-dependent UC-SF of NaYF₄: x mol%Nd³⁺@NaYF₄ UCNPs (xNd@Y; x = 2, 25, 50, 75, and 100). a–d** TEM images of xNd@Y UCNPs with different Nd³⁺ concentrations. **e** UCL enhancement factors for the Nd³⁺ emission at 588 nm in xNd@Y UCNPs with different Nd³⁺ concentrations, upon fs-pulsed laser excitation relative to that upon CW laser excitation with an equivalent power density at average (~1.10 kW cm⁻²). **f** Schematic of the establishment of macroscopic giant dipole in xNd@Y UCNPs at high and low Nd³⁺ concentrations. The black arrows denote the dipole phase and the orange curves represent the radiation field. **g** UC-SF decay curve of 50Nd@Y UCNPs upon 800-nm fs-pulsed laser excitation with a power density of 1.53 kW cm⁻². **h, i** Double logarithmic plots of the UCL intensities of the ⁴G₇/₂ → ⁴I₁₁/₂ and ⁴G₇/₂ → ⁴I₁₃/₂ transitions of Nd³⁺ at 588 and 656 nm versus the excitation power density (P) for (**h**) UC-SF and (**i**) normal UCL of 50Nd@Y UCNPs, showing different power dependence.

emission turned into a bi-exponential decay consisting of a fast component of single-pulse UC-SF (2.6 ns) and a slow component (>120 ns) originating from the propagation effect (Fig. 4b, c). This suggests that the emission of Nd³⁺ changed from pure UC-SF to the oscillatory regime by increasing the emitting sample length, wherein the oscillatory signal was evidently enhanced upon increasing the excitation power (Fig. 4d–g). Such excitation power dependence of oscillatory SF was also observed in other SF medium such as perovskite nanocrystal superlattices and hybrid perovskite films[24,39]. Specifically, through synergistic modulation of the excitation power and spot diameter, two periodical oscillatory peaks can be explicitly observed after the main UC-SF decay of Nd³⁺ (Fig. 4h), with a time interval of ~40 ns. These results provide solid evidence for the ultrafast UC-SF of Nd³⁺ with tunable lifetimes in either small or large active volumes of the Nd³⁺-concentrated system. The ultrafast UC-SF with a decay time on the ns scale provide an ideal solution to suppress the tailing effect associated with the μs−ms long lifetime of Ln³⁺ during the fast-scanning imaging (Supplementary Fig. 17), which is highly desirable for high-speed super-resolution bioimaging.

## Discussion

To conclude, we have demonstrated ultrafast UC-SF with a sub-2.5 ns lifetime in Nd³⁺-concentrated UCNPs. Specifically, the close proximity of Nd³⁺ enabled strong dipole-dipole correlations under fs-pulsed laser excitation, resulting a record-large number (N = 912) of coherently coupled dipoles and the consequent UC-SF with an ultrashort lifetime of sub-2.5 ns that is three orders of magnitude shorter than that

(2.28 μs) of normal UCL lifetime. The demonstration of giant coherence and ultrashort lifetime manipulation of Ln³⁺-doped UCNPs from the μs−ms scale to sub-ns represents a breakthrough in the development of UCNPs, as it enables the utilization of UCL of Ln³⁺ in ultrafast photonics that had long been sought after but restricted by the parity-forbidden 4 f → 4 f transitions. The fundamental understanding of UC-SF of Ln³⁺ lays also a foundation for the exploration of efficient and ultrafast UC materials toward a myriad of potential applications such as high-speed super-resolution bioimaging, quantum optics, and solid-state single-photon emitters.

## Methods

### Chemicals and materials

Nd(CH₃COO)₃·4H₂O (99.9%), Y(CH₃COO)₃·4H₂O (99.9%), Y(NO₃)₃·6H₂O (99.9%), Nd(NO₃)₃·6H₂O (99.9%), Yb(CH₃COO)₃·4H₂O (99.9%), Er(CH₃COO)₃·4H₂O (99.9%), ethylenediamine tetraacetic acid disodium salt dehydrate (EDTA-2Na, >99%), NaOH, NH₄F, NaF, oleic acid (OA), and 1-octadecene (ODE, 90%) were purchased from Sigma-Aldrich (China). NaHF₂ (98%) and aqueous suspension of polystyrene (PS) microspheres (MSs) (diameter 3.0−3.9 μm, 5 wt%) were purchased from Aladdin (Shanghai, China). Ethanol, butanol, and chloroform were purchased from Sinopharm Chemical Reagent Co. (China). All chemicals were used as received without further purification.

### Synthesis of NaYF₄: x mol%Nd³⁺ UCNPs

Monodisperse NaYF₄: x%Nd³⁺ (x = 2, 25, 50, 75, 100) UCNPs were synthesized through a high-temperature coprecipitation method by using

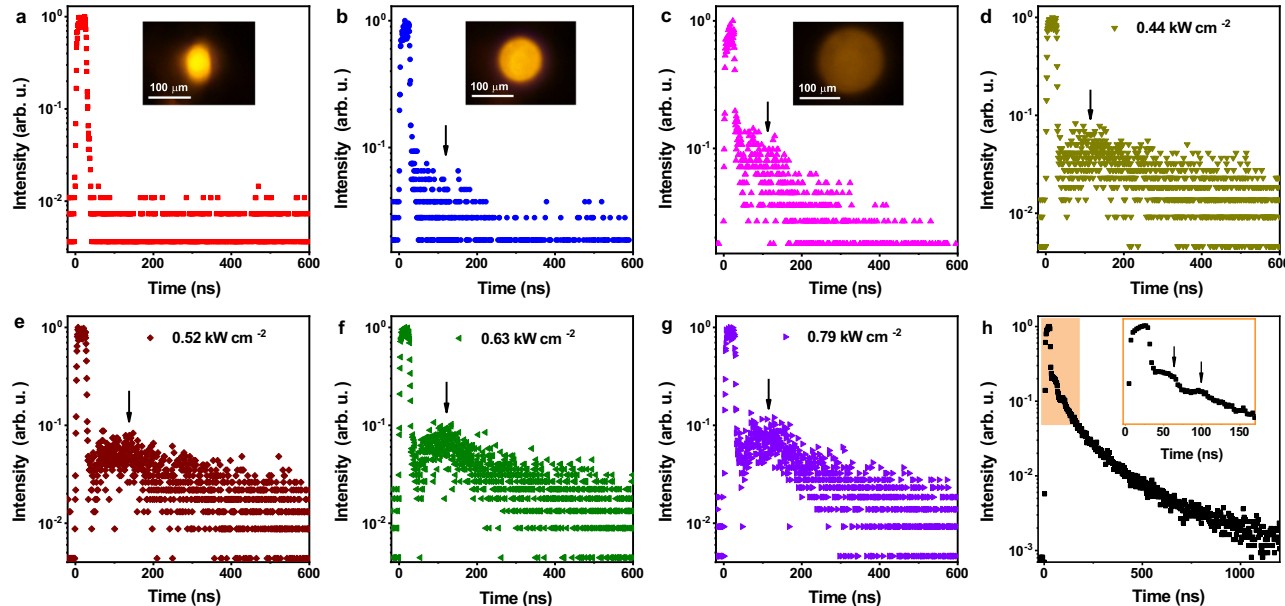

**Fig. 4 | Burnham-Chiao ringing of UC-SF in 100Nd@Y UCNPs.** UC-SF decay curves ($\lambda_{em}$ = 588 nm) of 100Nd@Y UCNPs upon fs-pulsed laser excitation at 800 nm with a power of 1.4 mW and an excitation spot diameter of **(a)** 56.3, **(b)** 105.7, and **(c)** 163.0 μm. The insets show the microscopic UC-SF images of the UCNP assemblies at the spot. UC-SF decay curves ($\lambda_{em}$ = 588 nm) of 100Nd@Y UCNPs at the condition of large emitting sample length upon 800-nm fs-pulsed laser excitation with a power density of **(d)** 0.44, **(e)** 0.52, **(f)** 0.63, and **(g)** 0.79 kW cm$^{-2}$. **h** UC-SF decay curve ($\lambda_{em}$ = 588 nm) of 100Nd@Y UCNPs upon 800-nm fs-pulsed laser excitation with a power density of 0.98 kW cm$^{-2}$ and an excitation spot diameter of 247.5 μm. The enlarged curve in the inset shows two periodical oscillatory peaks with a time interval of ~40 ns.

NaHF$_2$ as the sodium and fluoride sources. In a typical synthesis of NaYF$_4$: 2%Nd$^{3+}$ UCNPs, 0.02 mmol of Nd(CH$_3$COO)$_3$·4H$_2$O and 0.98 mmol of Y(CH$_3$COO)$_3$·4H$_2$O were mixed with 6 mL of OA and 15 mL of ODE in a 100 mL three-neck round-bottom flask. The mixture was heated to 180 °C under a N$_2$ flow with constant stirring for 30 min to form a clear solution. After cooling down to 80 °C, 2 mmol of NaHF$_2$ was added and the solution was heated to 260 °C with constant stirring for 30 min. Thereafter, the resulting solution was heated to 295 °C under a N$_2$ flow with vigorous stirring for 90 min, and then cooled down to RT. The obtained UCNPs were precipitated by the addition of 20 mL of ethanol, collected by centrifugation, washed with ethanol for three times, and finally redispersed in cyclohexane. NaYF$_4$: $x$%Nd$^{3+}$ UCNPs with different Nd$^{3+}$ concentrations were synthesized by adjusting the metal precursors (total amount of Nd + Y: 1 mmol) under otherwise identical conditions.

### Synthesis of NaYF$_4$: $x$ mol%Nd$^{3+}$@NaYF$_4$ ($x$Nd@Y) core-shell UCNPs

Briefly, 0.5 mmol of Y(CH$_3$COO)$_3$·4H$_2$O was mixed with 6 mL of OA and 15 mL of ODE in a 100 mL three-necked round-bottom flask. The mixture was heated to 160 °C under a N$_2$ flow with constant stirring for 30 min to form a clear solution. After cooling down to RT, 1 mmol of NaYF$_4$: $x$%Nd$^{3+}$ core-only UCNPs in 4 mL cyclohexane was added and the solution was heated to 80 °C for 45 min to remove cyclohexane. Then, 1 mmol of NaHF$_2$ was added and the solution was heated to 260 °C under a N$_2$ flow. Thereafter, the resulting solution was heated to 295 °C under a N$_2$ flow with vigorous stirring for 90 min, and then cooled down to RT. The obtained $x$Nd@Y core-shell UCNPs were precipitated by the addition of 20 mL of ethanol, collected by centrifugation, washed with ethanol three times, and finally redispersed in cyclohexane.

### Synthesis of large-size NaNdF$_4$@NaYF$_4$ core-shell UCNPs

Large-size NaNdF$_4$@NaYF$_4$ core-shell UCNPs were synthesized through a two-step hydrothermal method. Firstly, NaNdF$_4$ core-only

UCNPs were synthesized. In a typical synthesis, EDTA-2Na (2 mmol) was dissolved in 15 mL of distilled water followed by the addition of 2 mL aqueous solution of Nd(NO$_3$)$_3$ (0.5 M). The mixed solution was stirred for 40 min and then 10 mL aqueous solution of NaF (1.5 M) was added under vigorous stirring. Upon adjusting the pH to 1 and aging for 20 min, the mixture was transferred to a 50 mL Teflon-lined autoclave and allowed for reaction at 180 °C for 24 h. After cooling to RT, the precipitates were collected by centrifugation, washed with ethanol and distilled water several times, and finally re-dispersed in 5 mL of distilled water. For synthesizing NaNdF$_4$@NaYF$_4$ core-shell UCNPs, EDTA-2Na (2 mmol) was dissolved in 15 mL of distilled water followed by the addition of 1 mL aqueous solution of Y(NO$_3$)$_3$ (0.5 M). The mixed solution was stirred for 40 min and then 5 mL aqueous solution of NaF (1.5 M) was added under vigorous stirring. Subsequently, 5 mL aqueous solution of NaNdF$_4$ core-only UCNPs was added. Upon adjusting the pH to 1 and aging for 20 min, the mixture was transferred to a 50 mL Teflon-lined autoclave and allowed for reaction at 180 °C for 24 h. After cooling to RT, the precipitates were collected by centrifugation, washed with ethanol and distilled water several times, and finally re-dispersed in ethanol.

### Synthesis of NaYF$_4$: Yb/Er@NaYF$_4$ core-shell UCNPs

NaYF$_4$: Yb/Er@NaYF$_4$ core-shell UCNPs were synthesized through a high-temperature coprecipitation method. Firstly, NaYF$_4$: Yb/Er (18 mol%Yb$^{3+}$/2 mol%Er$^{3+}$) core-only UCNPs were synthesized. In a typical synthesis, 0.8 mmol of Y(CH$_3$COO)$_3$·4H$_2$O, 0.02 mmol of Er(CH$_3$COO)$_3$·4H$_2$O, and 0.18 mmol of Yb(CH$_3$COO)$_3$·4H$_2$O were mixed with 6 mL of OA and 15 mL of ODE in a 100 mL three-neck round-bottom flask. The mixture was heated to 160 °C under an N$_2$ flow with constant stirring for 30 min to form a clear solution. After cooling down to RT, 10 mL of methanol solution containing 4 mmol of NH$_4$F and 2.5 mmol of NaOH was added and the solution was stirred at 70 °C for 1 h to remove methanol. After methanol was evaporated, the resulting solution was heated to 300 °C under a N$_2$ flow with vigorous stirring for 1 h, and then cooled down to RT. The

obtained UCNPs were precipitated by the addition of 20 mL of ethanol, collected by centrifugation, washed with ethanol three times, and finally redispersed in cyclohexane. For synthesizing NaYF$_4$: Yb/Er@NaYF$_4$ core-shell UCNPs, 1 mmol of Y(CH$_3$COO)$_3$·4H$_2$O was mixed with 6 mL of OA and 15 mL of ODE in a 100 mL three-necked round-bottom flask. The mixture was heated to 160 °C under an N$_2$ flow with constant stirring for 30 min to form a clear solution. After cooling down to RT, 1 mmol of NaYF$_4$: Yb/Er core-only UCNPs in 5 mL cyclohexane was added and the solution was heated to 80 °C for 45 min to remove cyclohexane. Then, 10 mL of methanol solution containing 4 mmol of NH$_4$F and 2.5 mmol of NaOH was added and the solution was stirred at 70 °C for 1 h to remove methanol. After methanol was evaporated, the resulting solution was heated to 300 °C under an N$_2$ flow with vigorous stirring for 1 h, and then cooled down to RT. The obtained core-shell UCNPs were precipitated by the addition of 20 mL of ethanol, collected by centrifugation, washed with ethanol for three times, and finally redispersed in cyclohexane.

### Fabrication of UCMSs
Two different kinds of UCMSs were synthesized by coating NaNdF$_4$@NaYF$_4$ (100Nd@Y) and NaYF$_4$: Yb/Er@NaYF$_4$ (2Er@Y) core-shell UCNPs on the surface of PSMSs according to the procedure reported by Liu et al.[41] Specifically, 10 μL aqueous suspension of PSMSs (5 wt%) were added to the mixture of butanol (137 μL) and chloroform (12 μL). Thereafter, 5 μL cyclohexane solution of 100Nd@Y and 2Er@Y UCNPs (1 M) was added to the above mixture, respectively, vortexed, sonicated for 5 s, and then incubated at RT for 4 h. The resulting UCMSs were collected by centrifugation, washed with ethanol and hexane several times, and finally re-dispersed in ethanol.

### Characterizations
Powder XRD patterns of the UCNPs were collected with an X-ray diffractometer (MiniFlex2, Rigaku) using Cu Kα1 radiation (λ = 0.154187 nm). TEM measurements including the low- and high-resolution TEM, high-angle annular dark-field scanning TEM (HAADF-STEM), and EDX element mapping were performed on a TECNAI G2 F20 TEM. The scanning electron microscopy (SEM) measurements were performed by using a JSM-6700F SEM. Normal UCL lifetimes were measured on the FLS980 spectrometer (Edinburgh) equipped with a tunable midband Optical Parametric Oscillator (OPO) pulsed laser as the excitation source (410–2400 nm, 10 Hz, pulse width ≤5 ns, Vibrant 355II, OPOTEK). All the spectral data were recorded at RT and corrected for the spectral response of the spectrometer.

### UC-SF measurements
UC-SF spectra were measured by using UCNP film samples, prepared by drop casting a cyclohexane solution of the UCNPs (-0.5 M) onto a 40 × 25 mm glass coverslip. A customized inverted confocal microscope (Nikon, Ti-U) was used for the UC-SF measurements (Supplementary Fig. S4), which was equipped with an 808-nm CW diode laser (2 W, Changchun New Industries Optoelectronics Tech Co. Ltd) and an 800-nm fs-pulsed laser as two independent excitation sources. The fs-pulsed laser was generated by a regeneratively amplified femtosecond Ti: sapphire laser system (800 nm, 1000 Hz, pulse energy of 4 mJ, pulse width of 120 fs, Spitfire Pro-FIKXP, Spectra-Physics), which was seeded by a femtosecond Ti-sapphire oscillator (80 MHz, pulse width of 70 fs, 800 nm, Maitai XF-1, Spectra-Physics). Normal UCL spectra were measured upon excitation with the 808-nm CW diode laser. The excitation laser passed through a 750 nm short-pass dichroic mirror (Thorlabs, DMSP750R) and then was focused with a microscope objective lens (10 × 0.25 NA, Nikon) to the sample. The emission of the sample in the spectral range of 380–710 nm was collected by the same objective, spectrally filtered using a 750 nm short-pass filter (Thorlabs, FESH0750), and then captured by the spectrometer (Acton, SpectraPro-2300) or the photomultiplier tube (PMT) for spectroscopic

analysis. For power dependence measurements, a continuously variable, reflective neutral density filter wheel (Thorlabs) was inserted into the laser beam path for power selection. Powers were simultaneously recorded by a Thorlabs power meter. Average excitation power densities were calculated based on the measured laser powers and the 1/e$^2$ area for the employed excitation wavelength. For time-resolved UC-SF measurements, a time-correlated single-photon counter (TCSPC, PCS 900) was used to tag photon arrival times of collected luminescence with respect to the laser shutoff trigger event.

### Microscopic imaging of UCMSs
For microscopic imaging of UCMSs, a dilute ethanol dispersion of 100Nd@Y and 2Er@Y UCMSs (-0.1 wt%) was drop-casted onto a 40 × 25 mm glass coverslip, respectively. The imaging was performed on a two-photon excitation microscope (A1MP, Nikon) equipped with an 800-nm fs-pulsed laser, and the UCL signal was collected in the green (500–550 nm) and red channel (570–620 nm), respectively. The fs-pulsed laser was generated by a femtosecond Ti-sapphire oscillator (80 MHz, pulse width of 70 fs, 800 nm, Maitai XF-1, Spectra-Physics).

## Data availability
All data needed to evaluate the conclusions in the study are presented in the paper and/or the Supplementary Information. All data that support the findings within this paper are available from the corresponding authors upon request.

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

## Acknowledgements

This work was supported by the National Key R&D Program of China (grant number: 2022YFB3503700), the NSFC (grant numbers: 12174391, 12474418, 12104456, U22A20398, and 22135008), the Natural Science Foundation of Fujian Province (grant numbers: 2024I0040, 2024J010038, and 2023J05072), and the Self-deployment Project Research Program of Haixi Institutes, Chinese Academy of Sciences (Nos. CXZX–2022-GH10 and CXZX–2022-GS01).

## Author contributions

P.H. and Wei Z. conceived the project, designed the experiments, and wrote the manuscript. X.C. supervised the project, led the collaboration efforts, and revised the manuscript. M.Z. carried out the experiments and analyzed the data. X.S. designed the UC-SF measurement system and provided the technical support. R.Z., Wen Z., Z.S. and S.Z. helped with the UCNP synthesis and characterizations. All authors contributed to the general discussion and analysis of the manuscript.

## Competing interests

The authors declare no competing interests.
