## [Peer Review file · Nature Communications]

Ultrafast upconversion superfluorescence with a sub-2.5 ns lifetime at room temperature

Corresponding Author: Professor Xueyuan Chen

Version 0:

Reviewer comments:

Reviewer #1

(Remarks to the Author)

The paper titled "Ultrafast upconversion superfluorescence with a sub-2.5 ns lifetime at free space" presents interesting advancements in the field of nanophotonics. However, the improvements represent incremental advancements rather than groundbreaking shifts, as the use of lanthanide-doped nanoparticles for photon upconversion is a well-established technique. The essential material for superfluorescence is not new. The dependence on high-power femtosecond laser excitation limits broader applicability, as practical implementations often require more accessible light sources. Additionally, the research remains predominantly fundamental, with a less evident direct translational impact on applied sciences or industry, showing sub 2.5 ns can be uniquely used. While the paper surpasses previous records in the number of coherently coupled dipoles and decay rates, a more detailed comparative analysis with existing literature would provide a clearer picture of the actual novelty. Thus, the findings should be contextualized within the broader scope of ongoing advancements in the field.

Moreover, the paper should address a critical question: what is the quantum efficiency of this material, and is it indeed brighter than conventional upconversion nanoparticles (UCNPs)? The demonstration of practical advantages, such as enhanced brightness and efficacy in conventional multiphoton microscopy, is essential to validate the claims of superiority. Without clear evidence showing significantly improved quantum efficiency and brightness compared to traditional UCNPs, the practical impact and superiority of this material remain uncertain.

Furthermore, regarding the examples and demonstrations of use, it is crucial to assess whether the upconversion superfluorescence achieved is bright enough and effectively coupled with conventional multiphoton microscopy to demonstrate superiority over previous discoveries. The paper does not provide sufficient evidence to confirm that the brightness of the upconversion emission significantly surpasses previous methods when used with standard multiphoton microscopy systems. Without clear demonstrations showing enhanced imaging capabilities or other practical benefits in conventional setups, the claim of superiority remains unsubstantiated.

Reviewer #2

(Remarks to the Author)

This manuscript reports on bright and ultrafast (lifetime down to 2.5 ns) upconversion in $\text{NaYF}_4:x \text{ mol\% Nd}^{3+}@\text{NaYF}_4$ ($x = 2, 25, 50, 75, 100$) core-shell nanoparticles (UCNPs) at room temperature under free-space excitation. Upon fs-pulsed 800 nm laser excitation, the authors observe coherent upconversion emission with a two-fold increase in intensity and over a three-thousand-fold improvement in the radiative decay rate. The study further demonstrates the ability to manipulate the upconversion emission lifetime from microseconds to sub-nanoseconds by controlling excitation power and emitting sample length. This manipulation is accompanied by the characteristic superfluorescence signature of Burnham-Chiao ringing. These findings are intriguing and likely to capture the interest of the journal's readership. However, a deep discussion of the following points is necessary.

1) The authors argue that the observation of enhanced UCL intensity along with decay times on the ns scale may result from the burst of radiative transition rate of Nd^{3+} , signifying that the upconverted emission of $100\text{Nd}@\text{Y}$ under fs-pulsed laser excitation is not a normal UCL process. This assumption requires further justification (and support by literature data).

2) The authors must explain how the lifetime of the $\text{Nd}^{3+} 4G_{7/2}$ excited state is calculated. Furthermore, in Supplementary Figures S5 and S7 it is unclear the meaning of the full line. The excitation wavelength used in the experiments must be also reported.

3) On page 7 it is unjustified why "the distinct decay dynamics of the $\text{Nd}^{3+} 4G_{7/2} \rightarrow 4I_{11/2}$ and $4G_{7/2} \rightarrow 4I_{13/2}$ transitions

indicate the presence of two independent ensembles of coherently coupled dipoles, which might compete for the depletion of the shared upper level of $4G7/2$ (Fig. 1f).” Moreover, it is also vague why the two distinct lifetime values for the same excited state ($4G7/2$) confirms the establishment of the dipole-dipole correlations for UC-SF in $100\text{Nd}@Y$ UCNPs upon fs-pulsed laser excitation.

4) Correct the units for $\ln I$ in Figures 3h and 3i. The logarithmic function is a nondimensional quantity.

5) On page 8, the comparison between the reported upconversion superfluorescence (UC-SF) and data previously published (Ref. 25 of the manuscript) is speculative. Further data must be added to support the sentence “We speculate that the improved UC-SF properties achieved in our $100\text{Nd}@Y$ UCNPs lie in the use of small-sample SF system along with fs-pulsed laser excitation, which resulted in significantly enhanced radiation field and consequently increased number of coherently coupled dipoles as compared to those in large samples with ns-pulsed laser excitation.”

Reviewer #3

(Remarks to the Author)

The authors report on ultrafast UC-SF (Up-conversion Super-fluorescence) in Nd^{3+} -concentrated NaYF_4 : $x\%\text{Nd}^{3+}@Y$ core-shell UCNPs upon excitation with a 800-nm fs-pulsed laser at RT and free space. Owing to the strong coupling of Nd^{3+} under high radiation field provided by the fs-pulsed laser excitation, a huge number of coherently coupled dipoles of up to 912 was realized, resulting in three orders of magnitude improvement in the radiative decay rate of Nd^{3+} as compared to that of normal UCL, in parallel with a record-short lifetime of sub-2.5 ns that had never been achieved before. The study is very well conducted. The results are convincing, reproducible and include proofs of the many peculiarities expected for the manifestation of super-fluorescence. As such, the effects of the excitation power, Nd^{3+} concentration, and emitting sample length on UC-SF as well as its kinetics were investigated in detail. All the signatures of SF including the power-dependent build-up and decay times, fourth-order power dependence of the two-photon UC emission, and UC-SF oscillation (Burnham-Chiao ringing) were observed, providing solid evidence for UC-SF in the Nd^{3+} -concentrated system. The breakdown of radiative lifetime of Ln^{3+} from the μs – ms scale to sub-ns through UC-SF paves the straightforward way for Ln^{3+} luminescence in ultrafast optics toward various state-of-the-art applications.

I recommend the publication of the paper as is.

As a very minor improvement, please consider the reformulation of lines 19-20.

Version 1:

Reviewer comments:

Reviewer #1

(Remarks to the Author)

The authors have addressed many of the reviewers' comments, providing additional data, explanations, and clarifications.

However, their response falls short in demonstrating practical applications, with the current imaging demonstration being very preliminary and not quantified. Since room temperature superfluorescence for similar composition nanoparticles has been reported before, the novelty of this paper obviously lies in the small size of the nanoparticles, the use of a femtosecond laser, and the consequently shortened lifetime achieved. Given this context, practical cell imaging experiments are essential to demonstrate the advantages of these innovations.

The authors should conduct cell imaging experiments that clearly show how the shorter lifetime translates to improved imaging performance, reduced tailing effects compared to conventional UCNPs, and any improvements in resolution or speed of imaging. They should also demonstrate enhanced photostability compared to conventional dyes in multiphoton microscopes and determine whether these particles maintain their superfluorescence properties in aqueous phase.

These experiments are crucial to differentiate this work from previous room temperature SF reports and to establish its practical significance in the field. Additionally, given the small size of these nanoparticles, there's potential for easy modification and use in solution phase, which could be explored further. Without these more applied demonstrations, the practical superiority of these nanoparticles for real-world imaging applications remains somewhat theoretical, and the full impact of the shortened lifetime and small particle size on actual biomedical imaging applications cannot be fully assessed.

Reviewer #2

(Remarks to the Author)

The authors have effectively addressed the reviewers' comments, significantly enhancing the manuscript and making it suitable for publication. Additionally, I have been asked to comment on the application potential concerns raised by reviewer #1 and I think the experiments detailed in Supplementary Figure 17 adequately address these issues.

The imaging capabilities of upconversion superfluorescence (UC-SF) on a standard multiphoton microscopy system and its advantages over traditional UC luminescence are demonstrated in Supplementary Figure 17 of the revised Supplementary Information. The results show that UC microspheres, composed of $\text{NaNdF}_4@Y$ core-shell UCNPs coated on polystyrene, can be integrated with conventional multiphoton microscopy and provide sufficient brightness for high-quality imaging, especially for fast scanning applications. Moreover, the imaging performance of UC-SF is shown to be superior to traditional UC luminescence.

Reply to the Comments of Reviewer #1

General Comment:

The paper titled "Ultrafast upconversion superfluorescence with a sub-2.5 ns lifetime at free space" presents interesting advancements in the field of nanophotonics. However, the improvements represent incremental advancements rather than groundbreaking shifts, as the use of lanthanide-doped nanoparticles for photon upconversion is a well-established technique. The essential material for superfluorescence is not new. The dependence on high-power femtosecond laser excitation limits broader applicability, as practical implementations often require more accessible light sources. Additionally, the research remains predominantly fundamental, with a less evident direct translational impact on applied sciences or industry, showing sub 2.5 ns can be uniquely used. While the paper surpasses previous records in the number of coherently coupled dipoles and decay rates, a more detailed comparative analysis with existing literature would provide a clearer picture of the actual novelty. Thus, the findings should be contextualized within the broader scope of ongoing advancements in the field.

Moreover, the paper should address a critical question: what is the quantum efficiency of this material, and is it indeed brighter than conventional upconversion nanoparticles (UCNPs)? The demonstration of practical advantages, such as enhanced brightness and efficacy in conventional multiphoton microscopy, is essential to validate the claims of superiority. Without clear evidence showing significantly improved quantum efficiency and brightness compared to traditional UCNPs, the practical impact and superiority of this material remain uncertain.

Furthermore, regarding the examples and demonstrations of use, it is crucial to assess whether the upconversion superfluorescence achieved is bright enough and effectively coupled with conventional multiphoton microscopy to demonstrate superiority over previous discoveries. The paper does not provide sufficient evidence to confirm that the brightness of the upconversion emission significantly surpasses previous methods when used with standard multiphoton microscopy systems. Without clear demonstrations showing enhanced imaging capabilities or other practical benefits in conventional setups, the claim of superiority remains unsubstantiated.

Response:

We greatly appreciate the reviewer for his/her critical reading and helpful suggestions to improve the quality of our manuscript. We have carefully considered the reviewer's comments and revised our manuscript to address his/her concerns about the significance and novelty of our work. The significance and novelty of our work were not explicitly stressed out perhaps because of our unclear writing and data presentation in the original manuscript. In the revised manuscript, we have modified the text discussion and supplemented more experimental data including the demonstration of the superiority of upconversion (UC) superfluorescence (SF) compared to conventional UC luminescence (UCL) and the imaging of the UC-SF nanoparticles on the standard multiphoton microscopy, in an effort to strengthen the significance of our work.

To justify the significance and novelty of our findings, please allow us to briefly introduce the research background of our work. Photon upconversion through lanthanide (Ln^{3+})-doped nanoparticles is of great significance for various biomedical and nanophotonic applications. However, current development of upconversion nanoparticles (UCNPs) is hindered by the low absorption and emission efficiencies and the long radiative lifetimes (on the μs – ms scale) due to the parity-forbidden $4f \rightarrow 4f$ transition of Ln^{3+} ions, which restrict their applications in many advanced technologies such as time-dependent nanophotonics and ultrafast optics. SF is a quantum optical phenomenon in which an ensemble of emitters is coherently coupled to generate a short but intense burst of light. Recently, Lim and Han *et al.* discovered the room-temperature anti-Stokes-like UC-SF with a lifetime of 46 ns in Nd^{3+} -enriched UCNPs (*Nat. Photon.* **2022**, *16*, 737-742). However, the number of coherently coupled dipoles (N) reported was only 11, which is not ideal for the collective SF, because the SF emission intensity $I \propto N^2$ and the radiative decay time of SF $\tau_{\text{SF}} \propto \tau_{\text{SE}}/N$ (where τ_{SE} is the spontaneous decay time). Moreover, the fundamental photophysics of UC-SF including the UC-SF dynamics, the excitation power dependence, and the emitting sample length dependence remains unexplored.

The novelty of our work lies in two aspects that (1) we have reported **ultrafast UC-SF with a huge number ($N = 912$) of correlated dipoles and a record-short lifetime of sub-2.5 ns** in lanthanide-doped nanoparticles, which has not been achieved before; and (2) we have investigated the fundamental photophysics of UC-SF and **realized the lifetime manipulation of upconversion emission in a wide range from μs to sub-ns** through the control of excitation power and emitting sample length, which breaks through the limitations of UCNPs associated with the parity-forbidden $4f \rightarrow 4f$ transition of Ln^{3+} .

The following is the point-to-point response to the reviewer's comments.

Comment #1:

The paper titled "Ultrafast upconversion superfluorescence with a sub-2.5 ns lifetime at free space" presents interesting advancements in the field of nanophotonics. However, the improvements represent incremental advancements rather than groundbreaking shifts, as the use of lanthanide-doped nanoparticles for photon upconversion is a well-established technique. The essential material for superfluorescence is not new.

Response:

We greatly appreciate the reviewer for his/her positive feedback on our manuscript, noting that our work presents "interesting advancements in the field of nanophotonics". Meanwhile, we are sorry that the novelty of the central finding of our work, ultrafast UC-SF with a huge number ($N = 912$) of correlated dipoles and a record-short lifetime of sub-2.5 ns in Ln^{3+} -doped nanoparticles, was not explicitly stressed out in the original manuscript. It is true that Ln^{3+} -doped nanoparticles for photon UC is a well-established technique, however, the concept of UC-SF is brand new. In contrast to conventional UCL based on spontaneous emission (SE) from the uncorrelated dipoles of individual Ln^{3+} , UC-SF of Ln^{3+} -doped nanoparticles is a quantum optical phenomenon, in which the photo-excited dipoles of all Ln^{3+} ions in a single nanoparticle are coherently coupled and aligned in phase, appearing like a macroscopic giant

dipole to give rise to a short but intense burst of light. Therefore, UC-SF of Ln^{3+} -doped nanoparticles is drastically different from the well-established UCL, which provides an ideal solution to overcome the limitations of Ln^{3+} ions with respect to the low emission efficiency and long radiative lifetimes (on the μs – ms scale) restricted by the parity-forbidden $4f \rightarrow 4f$ transition.

To the best of our knowledge, there is only one paper reporting UC-SF in Ln^{3+} -doped nanoparticles in the literature (Ref. 25: Lim & Han *et al.*, *Nat. Photon.* **2022**, *16*, 737-742). In Lim & Han's work, the essential material is $\text{NaYF}_4:\text{Yb,Er}@/\text{NaYF}_4:\text{Yb}@/\text{NaNdF}_4:\text{Yb}$ core/shell/shell UCNPs with a large size of ~ 270 nm, and the excitation source for UC-SF is 800-nm ns-pulsed laser. However, the large-size UCNPs of ~ 270 nm and ns-pulsed laser excitation are not suitable for pure UC-SF, which yielded UC-SF with only a small number ($N = 11$) of coherently coupled dipoles and a lifetime of 46 ns. By contrast, the essential material in our study is $\text{NaNdF}_4@/\text{NaYF}_4$ with a size of ~ 54.9 nm, which was demonstrated to be an ideal small-sample SF system for pure UC-SF of Nd^{3+} at 588 nm and 656 nm. Moreover, we used the 800-nm fs-pulsed laser instead of the ns-pulsed laser as the excitation source to enhance the radiation field for phase synchronization of the dipoles, which resulted in remarkably improved UC-SF properties, in parallel with three orders of magnitude enhancement in the radiative decay rate, a record-short lifetime of sub-2.5 ns, and a huge number of coherently coupled dipoles of 912. In addition, our study delved into the fundamental photophysics of UC-SF, including the physical mechanism of the initial quantum stages, the emitting sample length dependence, and the tunability of the Burnham-Chiao ringing behavior, which is of vital importance for the exploration of UC-SF toward practical applications in the future but had not been addressed in Lim & Han's work.

To validate the superiority and strengthen the novelty of our work, we have carried out additional experiments and compared the results with those in the literature. See also the details in the following responses.

Comment #2:

The dependence on high-power femtosecond laser excitation limits broader applicability, as practical implementations often require more accessible light sources.

Response:

Thank you for your comment. We agree that practical implementations often require more accessible light sources. Although the fs-pulsed laser is not as easily available as continuous-wave (CW) diode laser due to its high cost, it had been widely applied in diverse fields such as multiphoton absorption of organic dyes and quantum dots, SF of perovskite nanocrystals, and super-resolution bioimaging (F. Helmchen *et al.*, *Nat. Methods* **2005**, *2*, 932-940; G. Raino *et al.*, *Nature* **2018**, *563*, 671-675; G. Findik *et al.*, *Nat. Photon.* **2021**, *15*, 676-680; M. Biliroglu *et al.*, *Nat. Photon.* **2022**, *16*, 324-329; M. Weber *et al.*, *Nat. Biotechnol.* **2023**, *41*, 569-576). We used the fs-pulsed laser as the excitation because it can provide a high peak power density, which is essential to boost the radiation field for the phase synchronization, resulting in UC-SF with a huge number of coherently coupled dipoles and an ultrafast decay rate. We believe that the cost of fs-pulsed laser will be significantly reduced with technological development and

become popular in the near future. In these regards, we don't think the use of fs-pulsed laser would be a problem.

Comment #3:

While the paper surpasses previous records in the number of coherently coupled dipoles and decay rates, a more detailed comparative analysis with existing literature would provide a clearer picture of the actual novelty. Thus, the findings should be contextualized within the broader scope of ongoing advancements in the field.

Response:

Many thanks for this valuable suggestion. To the best of our knowledge, there is only one paper reporting UC-SF in Ln^{3+} -doped nanoparticles in the literature (Ref. 25: Lim & Han *et al.*, *Nat. Photon.* **2022**, *16*, 737-742). Therefore, we compared the results about UC-SF with those reported by Lim & Han *et al.* Owing to the strong dipole-dipole correlations of Nd^{3+} in the small-sample UC-SF system under fs-pulsed laser excitation, we achieved a record-large number ($N = 912$) of coherently coupled dipoles and the consequent UC-SF with an ultrashort lifetime (τ_{SF}) of sub-2.5 ns, which represent a great shift in comparison with those ($N = 11$, $\tau_{\text{SF}} = 46$ ns) of the literature report by Lim & Han *et al.* We speculate that the improved UC-SF properties achieved in our study lie in the use of small-sample SF system along with fs-pulsed laser excitation, which resulted in significantly enhanced radiation field and consequently increased number of coherently coupled dipoles as compared to those in larger samples with ns-pulsed laser excitation.

To verify our speculation, we have synthesized $\text{NaNdF}_4@ \text{NaYF}_4$ UCNPs with a mean size of 233.5 nm (newly added data in Supplementary Fig. 7), which is comparable to that (~ 270 nm) of the literature report by Lim and Han *et al.* The newly synthesized UCNPs were labeled as L-100Nd@Y, and the UCNPs with a mean size of ~ 54.9 nm in our original manuscript were labeled as S-100Nd@Y. Upon fs-pulsed laser excitation at 800 nm with a power density of 2.09 kW cm^{-2} , L-100Nd@Y exhibited similar UC-SF of Nd^{3+} as that of S-100Nd@Y (newly added data in Supplementary Fig. 8a). However, the UC-SF decay time (τ_{SF}) of L-100Nd@Y (92.8 ns) was observed to be much longer than that (2.5 ns) of S-100Nd@Y (newly added data in Supplementary Fig. 8b), indicating smaller number of coherently coupled dipoles (N) for the ${}^4\text{G}_{7/2} \rightarrow {}^4\text{I}_{11/2}$ transition of Nd^{3+} in L-100Nd@Y than in S-100Nd@Y. To estimate the N value in L-100Nd@Y, we measured the normal UCL decay curve of Nd^{3+} ($\lambda_{\text{em}} = 588$ nm) in L-100Nd@Y (newly added data in Supplementary Fig. 8c), whereby the spontaneous decay time (τ_{SE}) from ${}^4\text{G}_{7/2}$ of Nd^{3+} was determined to be $2.05 \mu\text{s}$. According to the equation $\tau_{\text{SF}} \propto \tau_{\text{SE}}/N$, the number of coherently coupled dipoles for the ${}^4\text{G}_{7/2} \rightarrow {}^4\text{I}_{11/2}$ transition of Nd^{3+} in L-100Nd@Y was estimated to be ~ 23 , which is much lower than that (~ 912) in S-100Nd@Y. Nonetheless, it is significantly improved in comparison with that ($N = 11$) reported by Lim and Han *et al.* upon excitation with ns-pulsed laser. These results provide solid evidence for our speculation that the small-sample SF system along with fs-pulsed laser excitation is indeed suitable for achieving pure UC-SF with an ultrafast decay rate. The new experimental data and text discussion have been included as Supplementary Figs. 7 and 8 in the revised Supplementary Information.

Newly added data in the Supplementary Information:

Supplementary Figure 7. (a) XRD pattern of large-size $\text{NaNdF}_4@ \text{NaYF}_4$ (L-100Nd@Y) UCNPs. (b) and (c) Scanning electron microscopy (SEM) images of L-100Nd@Y UCNPs. (d) Size distribution of the UCNPs obtained by randomly calculating 100 particles in the SEM image.

Supplementary Figure 8. (a) UC-SF spectra and (b) decay curves ($\lambda_{em} = 588 \text{ nm}$) of the large-size ($\sim 233.5 \text{ nm}$, L-100Nd@Y) and small-size ($\sim 54.9 \text{ nm}$, S-100Nd@Y) $\text{NaNdF}_4@ \text{NaYF}_4$ UCNPs, upon fs-pulsed laser excitation at 800 nm with a power density of 2.09 kW cm^{-2} . (c) Normal UCL decay curve of L-100Nd@Y UCNPs by monitoring the ${}^4G_{7/2} \rightarrow {}^4I_{11/2}$ emission of Nd^{3+} at 588 nm upon excitation with a ns-pulsed laser at 808 nm (10 Hz , pulse width $\leq 5 \text{ ns}$). The UC-SF decay time (τ_{SF}) of L-100Nd@Y was determined to be 92.8 ns , which is much longer than that (2.5 ns) of S-100Nd@Y, indicating smaller number of coherently coupled dipoles (N) for the ${}^4G_{7/2} \rightarrow {}^4I_{11/2}$ transition of Nd^{3+} in L-100Nd@Y than in S-100Nd@Y. By single-exponential fitting to the decay curve in (c), the spontaneous decay time (τ_{SE}) from ${}^4G_{7/2}$ of Nd^{3+} in L-100Nd@Y was derived to be $2.05 \mu\text{s}$. According to the equation $\tau_{SF} \propto \tau_{SE}/N$, the number of coherently coupled dipoles (N) for the ${}^4G_{7/2} \rightarrow {}^4I_{11/2}$ transition of Nd^{3+} in L-100Nd@Y was estimated to be ~ 23 , which is much lower than that (~ 912) in S-100Nd@Y. Nonetheless, it is significantly improved in comparison with that ($N = 11$) reported by Lim and Han *et al.* under excitation with a high-repetition (1 k Hz) ns-pulsed laser. These observations demonstrate that the improved UC-SF properties achieved in our 100Nd@Y UCNPs lie in the use of small-sample SF system along with fs-pulsed laser

excitation, which resulted in significantly enhanced radiation field and consequently increased number of coherently coupled dipoles as compared to those in larger samples with ns-pulsed laser excitation.

Comment #4:

Moreover, the paper should address a critical question: what is the quantum efficiency of this material, and is it indeed brighter than conventional upconversion nanoparticles (UCNPs)? The demonstration of practical advantages, such as enhanced brightness and efficacy in conventional multiphoton microscopy, is essential to validate the claims of superiority. Without clear evidence showing significantly improved quantum efficiency and brightness compared to traditional UCNPs, the practical impact and superiority of this material remain uncertain.

Response:

Sorry for our unclear writing about the novelty of our work in the original manuscript. The main purpose of our work is not to sell the material of $\text{NaNdF}_4@NaYF_4$ (100Nd@Y), but to demonstrate the superiority of UC-SF of Ln^{3+} -doped UCNPs with an ultrashort lifetime on the ns scale, which breaks through the limitations of Ln^{3+} with respect to the low emission efficiency and the long radiative lifetimes (on the μs – ms scale) due to the parity-forbidden $4f \rightarrow 4f$ transition. We are not able to measure the quantum efficiency for UC-SF of 100Nd@Y on the microscopy due to the limitation of the instrument. It is also difficult to measure the UC quantum yield for the normal UCL of 100Nd@Y on the integrating sphere upon CW laser excitation, because Nd^{3+} is not a typical UC emitter due to the dense energy levels of Nd^{3+} that impose deleterious nonradiative energy losses through cross relaxation and energy migration among Nd^{3+} to the surface and lattice defects (M. Matulionyte *et al.*, *Chem. Rev.* **2023**, *123*, 515-554). We selected Nd^{3+} as the UC-SF emitter considering its strong absorption ($^4I_{9/2} \rightarrow ^4F_{5/2}$) at ~ 800 nm, which matches well the emission of the commercial fs-pulsed laser (800 nm, 1000 Hz, pulse energy of 4mJ, pulse width of 120 fs, Spitfire 407 Pro-FIKXP, Spectra-Physics). This facilitates us to delve into the fundamental photophysics of UC-SF, which is of vital importance for the exploration of UC-SF toward practical applications but remained unexplored so far.

To validate the superiority of the UC-SF nanoparticles compared to conventional UCNPs with normal UCL, we have carried out additional experiments including the imaging of 100Nd@Y UCNPs under the modes of UC-SF and normal UCL and the investigation of UC-SF in the benchmark $\text{NaYF}_4: \text{Yb/Er}$ UCNPs. For UC-SF measurements, a customized microscopic spectroscopy system was built based on the inverted confocal microscope (Nikon, Ti-U) and equipped with both CW and fs-pulsed laser as two independent excitation sources (Supplementary Fig. 4). Upon CW laser excitation at 808 nm with a power density of $\sim 1.10 \text{ kW cm}^{-2}$, 100Nd@Y UCNPs exhibited weak luminescence (normal UCL) with characteristic emission peaks at 526, 588, and 656 nm, corresponding to the electronic transitions of Nd^{3+} from $^4G_{7/2}$ to $^4I_{9/2}$, $^4I_{11/2}$, and $^4I_{13/2}$, respectively (Supplementary Fig. 5). By contrast, upon 800-nm fs-pulsed laser excitation with an equivalent power density at average (1.10 kW cm^{-2}), 100Nd@Y UCNPs displayed bright luminescence (microscopic images newly added in Fig. 1d), with 70 times enhancement in UCL intensity and the emergence of new emission peaks from high energy level of Nd^{3+} at 409 nm ($^2P_{1/2} \rightarrow ^4I_{9/2}$) and 449 nm ($^2P_{1/2} \rightarrow ^4I_{11/2}$). Such a giant UCL enhancement of Nd^{3+} is a result of the

collective emission of Nd^{3+} from the coherently coupled dipoles, namely, UC-SF as verified by the power-dependent and transient UCL measurements. These observations demonstrate unambiguously that UC-SF nanoparticles are superior to conventional UCNPs with normal UCL in terms of both efficiency and brightness. In the revised manuscript, we have supplemented additional experimental data and added more text discussion to demonstrate the superiority of UC-SF over normal UCL. See also text discussion newly added in lines 1-5 from the bottom of page 6 and lines 1-7 of page 7 and Fig. 1 in the revised manuscript, and Supplementary Fig. 5 in the revised Supplementary Information.

Very recently, we have also observed UC-SF in the benchmark NaYF_4 : Yb/Er UCNPs (~ 30 nm). As shown in Fig. Ra, upon fs-pulsed laser excitation at 980 nm with an average power density of ~ 170 W cm^{-2} , NaYF_4 : Yb/Er UCNPs exhibited bright luminescence with characteristic emission peaks at 522, 550, and 667 nm, corresponding to the electronic transitions of Er^{3+} from ${}^2\text{H}_{11/2}$, ${}^4\text{S}_{3/2}$, and ${}^4\text{F}_{9/2}$ to the ground state ${}^4\text{I}_{15/2}$, respectively. Owing to the very strong luminescence, the slit and integral time were set as small as possible to guarantee that the signal cannot exceed the response range of the instrument. For comparison, the UCNPs showed much weaker luminescence when 980-nm CW laser was used as the excitation source with an equivalent power density under otherwise identical conditions. Specifically, we found that the decay time of the ${}^4\text{F}_{9/2}$ state of Er^{3+} (667 nm) was shortened to ~ 102 ns upon fs-pulsed laser excitation, which is over three orders of magnitude shorter than its normal UCL lifetime (~ 130 μs) (Fig. Rb,c), indicating the occurrence of UC-SF. These results further confirm the superiority of UC-SF with significantly improved efficiency and brightness as compared to conventional UCL. This part is not included in the revised manuscript, because it is beyond the scope of current work. Moreover, the UC-SF measurement system coupled with 980-nm fs-pulsed laser is still under optimization and the UC-SF mechanism involving with energy transfer process is not clear now and requires further in-depth investigation.

Figure R. (a) Normal UCL and UC-SF spectra of NaYF_4 : Yb/Er UCNPs upon 980-nm CW and fs-pulsed laser excitation with an average power density of ~ 170 W cm^{-2} . The insets show the microscopic images for UC-SF and normal UCL of the UCNP assemblies. (b) UC-SF and (c) normal UCL decay curves of NaYF_4 : Yb/Er UCNPs by monitoring the ${}^4\text{F}_{9/2} \rightarrow {}^4\text{I}_{15/2}$ emission of Er^{3+} at 667 nm.

Revisions in the main text:

Lines 1-5 from the bottom of page 6 and lines 1-7 of page 7: We first measured the normal UCL spectrum

of 100Nd@Y UCNPs under CW laser excitation at 808 nm with a power density of $\sim 1.10 \text{ kW cm}^{-2}$. As shown Fig. 1d and Supplementary Fig. 5, the UCNPs exhibited weak luminescence with characteristic emission peaks at 526, 588, and 656 nm, corresponding to the electronic transitions of Nd^{3+} from $^4\text{G}_{7/2}$ to $^4\text{I}_{9/2}$, $^4\text{I}_{11/2}$, and $^4\text{I}_{13/2}$, respectively. The inefficient UCL of Nd^{3+} under CW laser excitation is not unexpected since Nd^{3+} is not a typical UC emitter due to the dense energy levels of Nd^{3+} that impose deleterious nonradiative energy losses through cross relaxation and energy migration among Nd^{3+} to the surface and lattice defects³¹. By contrast, upon 800-nm fs-pulsed laser excitation with an equivalent power density at average ($\sim 1.10 \text{ kW cm}^{-2}$), the UCNPs displayed bright luminescence (insets of Fig. 1d), with 70 times enhancement in UCL intensity and the emergence of new emission peaks from high energy level of Nd^{3+} at 409 nm ($^2\text{P}_{1/2} \rightarrow ^4\text{I}_{9/2}$) and 449 nm ($^2\text{P}_{1/2} \rightarrow ^4\text{I}_{11/2}$).

Revised Figure 1 in the main text:

Fig. 1. UC-SF of Nd^{3+} in $\text{NaNdF}_4@ \text{NaYF}_4$ (100Nd@Y) UCNPs. (a) Schematic of the build-up process of SF. The initially uncorrelated photo-excited dipoles with randomly distributed phases become correlated in phase through coherent coupling, forming a macroscopic giant dipole to generate SF. (b) TEM image and (c) EDX elemental mapping of 100Nd@Y UCNPs. (d) Normal UCL and UC-SF spectra of 100Nd@Y UCNPs upon 808-nm CW and 800-nm fs-pulsed laser excitation with an average power density of $\sim 1.10 \text{ kW cm}^{-2}$, respectively. The insets show the microscopic images for UC-SF and normal UCL of the UCNP assemblies. (e) Normal UCL and UC-SF decay curves of 100Nd@Y UCNPs by monitoring the $^4\text{G}_{7/2} \rightarrow ^4\text{I}_{11/2}$ and $^4\text{G}_{7/2} \rightarrow ^4\text{I}_{13/2}$ emissions of Nd^{3+} at 588 and 656 nm, respectively. The inset shows the enlarged UC-SF decay curves of Nd^{3+} at 588 and 656 nm. (f) Energy levels and electronic transitions of Nd^{3+} for normal UCL and UC-SF in 100Nd@Y UCNPs. GSA and ESA denote the ground-state absorption and excited-state absorption, respectively.

Reference newly added:

- 31 M. Matulionyte *et al.* The coming of age of neodymium: redefining its role in rare earth doped nanoparticles. *Chem. Rev.* **123**, 515-554 (2023).

Newly added data in the Supplementary Information:
Supplementary Figure 5. Normal upconversion luminescence (UCL) and UC-SF spectra of 100Nd@Y UCNPs upon 808-nm CW and 800-nm fs-pulsed laser excitation with an average power density of $\sim 1.10 \text{ kW cm}^{-2}$, respectively. The inset shows the enlarged UCL spectrum.

Comment #5:

Additionally, the research remains predominantly fundamental, with a less evident direct translational impact on applied sciences or industry, showing sub 2.5 ns can be uniquely used.....Furthermore, regarding the examples and demonstrations of use, it is crucial to assess whether the upconversion superfluorescence achieved is bright enough and effectively coupled with conventional multiphoton microscopy to demonstrate superiority over previous discoveries. The paper does not provide sufficient evidence to confirm that the brightness of the upconversion emission significantly surpasses previous methods when used with standard multiphoton microscopy systems. Without clear demonstrations showing enhanced imaging capabilities or other practical benefits in conventional setups, the claim of superiority remains unsubstantiated.

Response:

Thank you for this valuable suggestion. The imaging capability of UC-SF on the standard multiphoton microscopy system and its advantages over traditional UCL have been provided in Supplementary Fig. 17 in the revised Supplementary Information. To this regard, we have prepared UC microspheres (UCMSs) by coating $\text{NaNdF}_4@NaYF_4$ (100Nd@Y) and $\text{NaYF}_4: Yb/Er@NaYF_4$ (2Er@Y) core-shell UCNPs on the surface of polystyrene (PS) MSs ($\sim 3.5 \mu\text{m}$) (newly added data in Supplementary Figs. 17a,b). The as-prepared 100Nd@Y and 2Er@Y UCMSs can generate intense UC-SF from Nd^{3+} with an ultra-short decay time on the ns scale and normal UCL from Er^{3+} with a long decay time on the μs -ms scale, respectively, upon 800-nm fs-pulsed laser excitation. These UCMSs were then subjected to scanning imaging on the two-photon excitation microscopy (A1MP, Nikon) under excitation with an 800-

nm fs-pulsed laser. The results showed that UC-SF can be coupled well with conventional multiphoton microscopy and is bright enough for high-quality imaging, especially for fast scanning imaging. As shown in Supplementary Figs. 17c,d, when the pixel dwell time was set as 1.1 μ s, the images of 2Er@Y UCMSs suffered from an obvious tailing effect due to the long radiative lifetime of normal UCL, which caused the image distortion and deformation. For comparison, the images of 100Nd@Y UCMSs were clear and free from tailing effect because of the ultra-short decay time of UC-SF (Supplementary Figs. 17e,f). These results reveal the great potential of UC-SF for high-speed and high-resolution bioimaging. See also text discussion newly added in line 1 from the bottom of page 14 and lines 1-2 of page 15 in the revised manuscript and Supplementary Fig. 17 newly added in the revised Supplementary Information.

Text discussion newly added in the main text:

Line 1 from the bottom of page 14 and lines 1-2 of page 15: The ultrafast UC-SF with a decay time on the ns scale provide an ideal solution to suppress the tailing effect associated with the μ s–ms long lifetime of Ln^{3+} during the fast-scanning imaging (Supplementary Fig. 17), which is highly desirable for high-speed super-resolution bioimaging.

Newly added data in the Supplementary Information:

Supplementary Figure 17. SEM images of (a) 2Er@Y and (b) 100Nd@Y UCMSs. Laser scanning microscopy images of (c,d) 2Er@Y and (e,f) 100Nd@Y UCMSs, collected in the green (500-550 nm) and red channel (570-620 nm), respectively. Image dimensions: 1024 \times 1024 pixels; pixel size: 0.11 μ m; pixel dwell time: 1.1 μ s; acquisition time: 16.6 s. When the pixel dwell time was set as 1.1 μ s, the images of 2Er@Y UCMSs suffered from an obvious tailing effect due to the μ s–ms long radiative lifetime of normal UCL, which caused the image distortion and deformation. For comparison, the images of 100Nd@Y UCMSs were clear and free from tailing effect because of the ultra-short decay time (on the ns scale) of UC-SF.

To summarize, we have modified the text discussion and supplemented more experimental data to strengthen the novelty of this work. The significance and novelty of our work are also supported by Reviewer #2 who ranked our work as “intriguing and likely to capture the interest of the journal's readership” and Reviewer #3 who recommended publication of our manuscript as it is. We believe that, our finding is novel and significant as it presents a substantial advance with respect to previous work and could be a breakthrough in the development of UCNPs, which lays a foundation for the exploration of efficient and ultrafast UC materials toward a myriad of potential applications such as high-speed super-resolution bioimaging, quantum optics, and solid-state single-photon emitters. We sincerely hope the reviewer concurs after reading the revised manuscript and the above clarification.

Reply to the Comments of Reviewer #2

General Comment:

This manuscript reports on bright and ultrafast (lifetime down to 2.5 ns) upconversion in NaYF₄:x mol% Nd³⁺@NaYF₄ (x = 2, 25, 50, 75, 100) core-shell nanoparticles (UCNPs) at room temperature under free-space excitation. Upon fs-pulsed 800 nm laser excitation, the authors observe coherent upconversion emission with a two-fold increase in intensity and over a three-thousand-fold improvement in the radiative decay rate. The study further demonstrates the ability to manipulate the upconversion emission lifetime from microseconds to sub-nanoseconds by controlling excitation power and emitting sample length. This manipulation is accompanied by the characteristic superfluorescence signature of Burnham-Chiao ringing. These findings are intriguing and likely to capture the interest of the journal's readership. However, a deep discussion of the following points is necessary.

Response:

We greatly appreciate the reviewer for his/her positive comments on our manuscript and the efforts to improve the quality of our manuscript. We have carefully looked into all the helpful suggestions by the reviewer and made all the requested changes that have been reflected either in the text or in the Supplementary Information. A point-by-point response is noted below.

Comment #1:

The authors argue that the observation of enhanced UCL intensity along with decay times on the ns scale may result from the burst of radiative transition rate of Nd³⁺, signifying that the upconverted emission of 100Nd@Y under fs-pulsed laser excitation is not a normal UCL process. This assumption requires further justification (and support by literature data).

Response:

This assumption was made based on the following considerations. We found that the decay times of the ⁴G_{7/2} → ⁴I_{11/2} (588 nm) and ⁴G_{7/2} → ⁴I_{13/2} (656 nm) transitions of Nd³⁺ under fs-pulsed laser excitation were different and abnormally shortened to 10.7 and 24.8 ns, respectively (inset of Fig. 1e). These observations are in stark contrast to the normal UCL lifetimes of Nd³⁺ under ns-pulsed laser excitation, where Nd³⁺ displayed an identical UCL decay time (τ_{SE}) of 2.28 μ s at 588 and 656 nm (Fig. 1e). Generally, the decay times of the parity-forbidden 4f → 4f transitions of Ln³⁺ ions are on the μ s–ms range and the decay times of the UCL from the same emitting level should be identical, because each transition dipole shares the same deexcitation channels of the emitting level (J. C. Bünzli & S. Eliseeva, *Springer Berlin Heidelberg* **2011**, 7, 1-45). Additionally, in normal UCL, the enhancement of UCL intensity is usually accompanied by the lengthening of UCL lifetime due to the suppressed nonradiative relaxation, while the shortening of UCL lifetime indicates accelerated nonradiative relaxation of excited Ln³⁺ ions through energy transfer to the lattice or surface defects, which results in decreased UCL intensity (G. Y. Chen *et al.*, *Chem. Soc. Rev.* **2015**, 44, 1680-1713; N. J. J. Johnson *et al.*, *J. Am. Chem. Soc.* **2017**, 139, 3275–3282). Therefore, we deduced that the observation of remarkably enhanced UCL intensity along

with decay times on the ns scale may result from the burst of radiative transition rate of Nd^{3+} instead of the suppressed nonradiative relaxation, signifying that the upconverted emission of 100Nd@Y under fs-pulsed laser excitation is not a normal UCL process. To rationalize this argument, we have added more text discussion about the UCL enhancement along with the shortened decay times on the ns scale observed in 100Nd@Y under fs-pulsed laser excitation. Two papers discussing the relationship between the UCL intensity and lifetime have also been included as new references (Refs. 33 and 34). See also text discussion newly added in lines 14-20 of page 7 and Refs. 33 and 34 newly added in the revised manuscript.

Revisions in the main text:

Lines 14-20 of page 7: Additionally, in normal UCL, the enhancement of UCL intensity is usually accompanied by the lengthening of UCL lifetime due to the suppressed nonradiative relaxation, while the shortening of UCL lifetime indicates accelerated nonradiative relaxation of excited Ln^{3+} ions through energy transfer to the lattice or surface defects, which results in decreased UCL intensity^{33,34}. Hence, the observation of remarkably enhanced UCL intensity along with decay times on the ns scale may result from the burst of radiative transition rate of Nd^{3+} instead of the suppressed nonradiative relaxation, signifying that the upconverted emission of 100Nd@Y under fs-pulsed laser excitation is not a normal UCL process.

References newly added:

- 33 Chen, G. Y. *et al.* Light upconverting core-shell nanostructures: nanophotonic control for emerging applications. *Chem. Soc. Rev.* **44**, 1680-1713 (2015).
- 34 Johnson, N. J. J. *et al.* Direct evidence for coupled surface and concentration quenching dynamics in lanthanide-doped nanocrystals. *J. Am. Chem. Soc.* **139**, 3275–3282 (2017).

Comment #2:

The authors must explain how the lifetime of the $\text{Nd}^{3+} \ ^4G_{7/2}$ excited state is calculated. Furthermore, in Supplementary Figures 5 and 7 it is unclear the meaning of the full line. The excitation wavelength used in the experiments must be also reported.

Response:

Many thanks for this valuable suggestion. The UC-SF decay curves in Supplementary Figs. 6 and 10 (original Supplementary Figs. 5 and 7) were obtained upon fs-pulsed laser excitation at 800 nm with different power densities, and the full lines represent the fitting results to the decay and rise components of the decay curves, respectively. These important messages have been supplemented in the captions of Supplementary Figs. 6 and 10 in the revised Supplementary Information.

The UC-SF decay curves of Nd^{3+} ($\lambda_{\text{em}} = 588$ nm) consisted of the rise and decay components,

corresponding to the build-up and decay processes of UC-SF, respectively. The radiative decay times (τ_R) for UC-SF of Nd^{3+} were determined by single- or bi-exponential fitting to the decay component of the decay curves (Supplementary Fig. 6). At low excitation power densities (0.71–0.75 kW cm^{-2}), the emission of Nd^{3+} deviated from single-exponential decay, due to the mixing of the normal UCL of Nd^{3+} . In this case, the decay curves were fitted with a biexponential function:

$$I(t) = A_1 \exp\left(-\frac{t}{\tau_1}\right) + A_2 \exp\left(-\frac{t}{\tau_2}\right)$$

where $I(t)$ denotes the luminescence intensity as a function of time; A_1 and A_2 are the weight ratios for the lifetime components of τ_1 and τ_2 , respectively. The average decay time of Nd^{3+} was calculated by the following expression:

$$\tau_{ave} = \frac{A_1 \times \tau_1^2 + A_2 \times \tau_2^2}{A_1 \times \tau_1 + A_2 \times \tau_2}$$

At high excitation power densities (1.07–2.09 kW cm^{-2}), the emission of Nd^{3+} turned to pure UC-SF and the decay curves can be fitted with a single-exponential function, whereby the decay time of Nd^{3+} was derived. Based on the fitting results, the radiative decay time for UC-SF of Nd^{3+} was determined to decrease from 37.4 ns to 2.5 ns with the increasing excitation power density from 0.71 to 2.09 kW cm^{-2} , due to the increased number of coupled dipoles in the coherent state. Concurrently, the rise component of the decay curves (Supplementary Fig. 10) was fitted by a single-exponential function and the delay time was determined as 95% of the asymptotic value. The details for the calculation of the radiative decay time (τ_R) and delay time (τ_D) for UC-SF of Nd^{3+} from the ${}^4G_{7/2}$ state have been supplemented in the captions of Supplementary Figs. 6 and 10 in the revised Supplementary Information.

Revisions in the Supplementary Information:

Supplementary Figure 6. Power-dependent UC-SF decay curves ($\lambda_{em} = 588 \text{ nm}$) of $\text{NaNdF}_4@NaYF_4$

UCNPs upon fs-pulsed laser excitation at 800 nm, showing the decreased radiative decay time (τ_R) with the increasing excitation power density. The blue full lines represent the fitting curves to the decay component. At low excitation power densities (0.71–0.75 kW cm⁻²), the emission of Nd³⁺ deviated from single-exponential decay, due to the mixing of the normal UCL of Nd³⁺. In this case, the decay curves were fitted with a biexponential function:

$$I(t) = A_1 \exp\left(-\frac{t}{\tau_1}\right) + A_2 \exp\left(-\frac{t}{\tau_2}\right)$$

where $I(t)$ denotes the luminescence intensity as a function of time; A_1 and A_2 are the weight ratios for the lifetime components of τ_1 and τ_2 , respectively. The average decay time of Nd³⁺ was calculated by the following expression:

$$\tau_{ave} = \frac{A_1 \times \tau_1^2 + A_2 \times \tau_2^2}{A_1 \times \tau_1 + A_2 \times \tau_2}$$

At high excitation power densities (1.07–2.09 kW cm⁻²), the emission of Nd³⁺ turned to pure UC-SF and the decay curves can be fitted with a single-exponential function, whereby the decay time of Nd³⁺ was derived.

Supplementary Figure 10. Power-dependent UC-SF decay curves ($\lambda_{em} = 588$ nm) of NaNdF₄@NaYF₄ UCNPs at the initial stage upon fs-pulsed laser excitation at 800 nm, showing the decreased delay time (τ_D) with the increasing excitation power density. The blue full lines represent the single-exponential fitting to the rise component of the decay curves and the delay time was determined as 95% of the asymptotic value.

Comment #3:

On page 7 it is unjustified why “the distinct decay dynamics of the Nd³⁺ $^4G_{7/2} \rightarrow ^4I_{11/2}$ and $^4G_{7/2} \rightarrow ^4I_{13/2}$

transitions indicate the presence of two independent ensembles of coherently coupled dipoles, which might compete for the depletion of the shared upper level of ${}^4G_{7/2}$ (Fig. 1f).” Moreover, it is also vague why the two distinct lifetime values for the same excited state (${}^4G_{7/2}$) confirms the establishment of the dipole-dipole correlations for UC-SF in 100Nd@Y UCNPs upon fs-pulsed laser excitation.

Response:

Thank you for pointing out this important issue. Actually, Ln^{3+} -doped UCNPs are a multilevel SF system because of the abundant electronic transitions within a single Ln^{3+} ion. For SF in a multilevel system, the emission can occur successively on two cascading transitions at two different frequencies or two transitions with different frequencies sharing a common upper level, which can be in competition for the depletion of this level, due to the different frequencies or different polarizations of the radiation fields (M. Gross & S. Haroche, *Phys. Rep.* **1982**, 93, 301-396). The ${}^4G_{7/2} \rightarrow {}^4I_{11/2}$ and ${}^4G_{7/2} \rightarrow {}^4I_{13/2}$ transitions of Nd^{3+} are the Λ -type competing transitions, in which the excited state can decay to multiple ground states with different frequencies (S. J. Masson *et al.*, *PRX Quantum* **2024**, 5, 010344). For better clarity, we have added more text discussion about the implication of the distinct decay dynamics of the ${}^4G_{7/2} \rightarrow {}^4I_{11/2}$ and ${}^4G_{7/2} \rightarrow {}^4I_{13/2}$ transitions of Nd^{3+} . A paper about SF and superradiance in multilevel systems has also been included as a new reference (Ref. 35). See also text discussion newly added in lines 1-8 of page 8 and Ref. 35 newly added in the revised manuscript.

Text discussion newly added in the main text:

Lines 1-8 of page 8: Actually, Ln^{3+} -doped UCNPs are a multilevel SF system because of the abundant electronic transitions within a single Ln^{3+} ion. For SF in a multilevel system, the emission can occur successively on two cascading transitions at two different frequencies or two transitions with different frequencies sharing a common upper level, which can be in competition for the depletion of this level, due to the different frequencies or different polarizations of the radiation fields³⁰. The transitions from ${}^4G_{7/2}$ to ${}^4I_{11/2}$ and ${}^4I_{13/2}$ of Nd^{3+} are the Λ -type competing transitions, in which the excited state can decay to multiple ground states with different frequencies³⁵. Therefore, the distinct decay dynamics of the ${}^4G_{7/2} \rightarrow {}^4I_{11/2}$ and ${}^4G_{7/2} \rightarrow {}^4I_{13/2}$ transitions of Nd^{3+} indicates the presence of two independent ensembles of coherently coupled dipoles, which might compete for the depletion of the shared upper level of ${}^4G_{7/2}$ (Fig. 1f). This phenomenon further confirms the establishment of the dipole-dipole correlations for UC-SF in 100Nd@Y UCNPs upon fs-pulsed laser excitation.

Reference newly added:

35 Masson, S. J. *et al.* Dicke superradiance in ordered arrays of multilevel atoms. *PRX Quantum* **5**, 010344 (2024).

Comment #4:

Correct the units for $\ln I$ in Figures 3h and 3i. The logarithmic function is a nondimensional quantity.

Response:

Thank you for your kind reminding. We have corrected the units for $\ln I$ in Figs. 3h and 3i in the revised manuscript.

Revised Figure 3 in the main text:

Fig. 3. Nd³⁺ concentration-dependent UC-SF of NaYF₄: x mol%Nd³⁺@NaYF₄ UCNPs (xNd@Y; x = 2, 25, 50, 75, and 100). (a-d) TEM images of xNd@Y UCNPs with different Nd³⁺ concentrations. (e) UCL enhancement factors for the Nd³⁺ emission at 588 nm in xNd@Y UCNPs with different Nd³⁺ concentrations, upon fs-pulsed laser excitation relative to that upon CW laser excitation with an equivalent power density at average (~1.10 kW cm⁻²). (f) Schematic of the establishment of macroscopic giant dipole in xNd@Y UCNPs at high and low Nd³⁺ concentrations. The black arrows denote the dipole phase and the orange curves represent the radiation field. (g) UC-SF decay curve of 50Nd@Y UCNPs upon 800-nm fs-pulsed laser excitation with a power density of 1.53 kW cm⁻². (h, i) Double logarithmic plots of the UCL intensities of the ⁴G_{7/2} → ⁴I_{11/2} and ⁴G_{7/2} → ⁴I_{13/2} transitions of Nd³⁺ at 588 and 656 nm versus the excitation power density (P) for (h) UC-SF and (i) normal UCL of 50Nd@Y UCNPs, showing different power dependence.

Comment #5:

On page 8, the comparison between the reported upconversion superfluorescence (UC-SF) and data previously published (Ref. 25 of the manuscript) is speculative. Further data must be added to support the sentence “We speculate that the improved UC-SF properties achieved in our 100Nd@Y UCNPs lie in the use of small-sample SF system along with fs-pulsed laser excitation, which resulted in significantly enhanced radiation field and consequently increased number of coherently coupled dipoles as compared

to those in large samples with ns-pulsed laser excitation.”

Response:

Thank you for this valuable suggestion. To verify our speculation, we have synthesized $\text{NaNdF}_4@\text{NaYF}_4$ UCNPs with a mean size of 233.5 nm (newly added data in Supplementary Fig. 7), which is comparable to that (~ 270 nm) of the literature report by Lim and Han *et al.* (Ref. 25). The newly synthesized UCNPs were labeled as L-100Nd@Y, and the UCNPs with a mean size of ~ 54.9 nm in our original manuscript were labeled as S-100Nd@Y. Upon fs-pulsed laser excitation at 800 nm with a power density of 2.09 kW cm^{-2} , L-100Nd@Y exhibited similar UC-SF of Nd^{3+} as that of S-100Nd@Y (newly added data in Supplementary Fig. 8a). However, the UC-SF decay time (τ_{SF}) of L-100Nd@Y (92.8 ns) was much longer than that (2.5 ns) of S-100Nd@Y (newly added data in Supplementary Fig. 8b), indicating smaller number of coherently coupled dipoles (N) for the ${}^4\text{G}_{7/2} \rightarrow {}^4\text{I}_{11/2}$ transition of Nd^{3+} in L-100Nd@Y than in S-100Nd@Y. To estimate the N value in L-100Nd@Y, we measured the normal UCL decay curve of Nd^{3+} ($\lambda_{\text{em}} = 588 \text{ nm}$) in L-100Nd@Y upon excitation with a ns-pulsed laser at 808 nm (10 Hz, pulse width $\leq 5 \text{ ns}$) (newly added data in Supplementary Fig. 8c), whereby the spontaneous decay time (τ_{SE}) from ${}^4\text{G}_{7/2}$ of Nd^{3+} was determined to be $2.05 \mu\text{s}$. According to the equation $\tau_{\text{SF}} \propto \tau_{\text{SE}}/N$, the number of coherently coupled dipoles (N) for the ${}^4\text{G}_{7/2} \rightarrow {}^4\text{I}_{11/2}$ transition of Nd^{3+} in L-100Nd@Y was estimated to be ~ 23 , which is much lower than that (~ 912) in S-100Nd@Y. We are not able to measure the UC-SF of the samples upon 800-nm ns-pulsed laser excitation with a high repetition frequency (1k Hz), because it is not available in our lab. Nonetheless, the number of coherently coupled dipoles ($N = 23$) in L-100Nd@Y upon fs-pulsed laser excitation is significantly improved in comparison with that ($N = 11$) reported by Lim and Han *et al.* under excitation with a high-repetition (1k Hz) ns-pulsed laser. These observations demonstrate that the improved UC-SF properties achieved in our 100Nd@Y UCNPs lie in the use of small-sample SF system along with fs-pulsed laser excitation, which resulted in significantly enhanced radiation field and consequently increased number of coherently coupled dipoles as compared to those in larger samples with ns-pulsed laser excitation. The new experimental data and text discussion have been included as Supplementary Figs. 7 and 8 in the revised Supplementary Information.

Newly added data in the Supplementary Information:

Supplementary Figure 7. (a) XRD pattern of large-size $\text{NaNdF}_4@\text{NaYF}_4$ (L-100Nd@Y) UCNPs. (b)

and (c) Scanning electron microscopy (SEM) images of L-100Nd@Y UCNPs. (d) Size distribution of the UCNPs obtained by randomly calculating 100 particles in the SEM image.

Supplementary Figure 8. (a) UC-SF spectra and (b) decay curves ($\lambda_{em} = 588$ nm) of the large-size (~ 233.5 nm, L-100Nd@Y) and small-size (~ 54.9 nm, S-100Nd@Y) NaNdF₄@NaYF₄ UCNPs, upon fs-pulsed laser excitation at 800 nm with a power density of 2.09 kW cm⁻². (c) Normal UCL decay curve of L-100Nd@Y UCNPs by monitoring the ${}^4G_{7/2} \rightarrow {}^4I_{11/2}$ emission of Nd³⁺ at 588 nm upon excitation with a ns-pulsed laser at 808 nm (10 Hz, pulse width ≤ 5 ns). The UC-SF decay time (τ_{SF}) of L-100Nd@Y was determined to be 92.8 ns, which is much longer than that (2.5 ns) of S-100Nd@Y, indicating smaller number of coherently coupled dipoles (N) for the ${}^4G_{7/2} \rightarrow {}^4I_{11/2}$ transition of Nd³⁺ in L-100Nd@Y than in S-100Nd@Y. By single-exponential fitting to the decay curve in (c), the spontaneous decay time (τ_{SE}) from ${}^4G_{7/2}$ of Nd³⁺ in L-100Nd@Y was derived to be 2.05 μ s. According to the equation $\tau_{SF} \propto \tau_{SE}/N$, the number of coherently coupled dipoles (N) for the ${}^4G_{7/2} \rightarrow {}^4I_{11/2}$ transition of Nd³⁺ in L-100Nd@Y was estimated to be ~ 23 , which is much lower than that (~ 912) in S-100Nd@Y. Nonetheless, it is significantly improved in comparison with that ($N = 11$) reported by Lim and Han *et al.* under excitation with a high-repetition (1k Hz) ns-pulsed laser. These observations demonstrate that the improved UC-SF properties achieved in our 100Nd@Y UCNPs lie in the use of small-sample SF system along with fs-pulsed laser excitation, which resulted in significantly enhanced radiation field and consequently increased number of coherently coupled dipoles as compared to those in larger samples with ns-pulsed laser excitation.

Reply to the Comments of Reviewer #3

General Comment:

The authors report on ultrafast UC-SF (Up-conversion Super-fluorescence) in Nd^{3+} -concentrated NaYF_4 : $x\%\text{Nd}^{3+}$ @ NaYF_4 core-shell UCNPs upon excitation with a 800-nm fs-pulsed laser at RT and free space. Owing to the strong coupling of Nd^{3+} under high radiation field provided by the fs-pulsed laser excitation, a huge number of coherently coupled dipoles of up to 912 was realized, resulting in three orders of magnitude improvement in the radiative decay rate of Nd^{3+} as compared to that of normal UCL, in parallel with a record-short lifetime of sub-2.5 ns that had never been achieved before. The study is very well conducted. The results are convincing, reproducible and include proofs of the many peculiarities expected for the manifestation of super-fluorescence. As such, the effects of the excitation power, Nd^{3+} concentration, and emitting sample length on UC-SF as well as its kinetics were investigated in detail. All the signatures of SF including the power-dependent build-up and decay times, fourth-order power dependence of the two-photon UC emission, and UC-SF oscillation (Burnham-Chiao ringing) were observed, providing solid evidence for UC-SF in the Nd^{3+} -concentrated system. The breakdown of radiative lifetime of Ln^{3+} from the μs – ms scale to sub-ns through UC-SF paves the straightforward way for Ln^{3+} luminescence in ultrafast optics toward various state-of-the-art applications.

I recommend the publication of the paper as is.

Response:

We greatly appreciate the reviewer for his/her positive comments on our manuscript and the efforts to improve the quality of our manuscript. We have carefully looked into all the helpful suggestions by the reviewer and made all the requested changes that have been reflected in the text. A point-by-point response is noted below.

Comment #1:

As a very minor improvement, please consider the reformulation of lines 19-20.

Response:

The sentence in lines 19-20 has been refined as “Specifically, we achieve a huge number ($N = 912$) of correlated dipoles in Nd^{3+} -concentrated nanoparticles upon excitation with a fs-pulsed laser, resulting in collective coherent emission with two orders of magnitude amplification in intensity and more than three orders of magnitude improvement in the radiative decay rate”.

Reply to the Comments of Reviewer #1

Comments:

The authors have addressed many of the reviewers' comments, providing additional data, explanations, and clarifications.

However, their response falls short in demonstrating practical applications, with the current imaging demonstration being very preliminary and not quantified. Since room temperature superfluorescence for similar composition nanoparticles has been reported before, the novelty of this paper obviously lies in the small size of the nanoparticles, the use of a femtosecond laser, and the consequently shortened lifetime achieved. Given this context, practical cell imaging experiments are essential to demonstrate the advantages of these innovations.

The authors should conduct cell imaging experiments that clearly show how the shorter lifetime translates to improved imaging performance, reduced tailing effects compared to conventional UCNPs, and any improvements in resolution or speed of imaging. They should also demonstrate enhanced photostability compared to conventional dyes in multiphoton microscopes and determine whether these particles maintain their superfluorescence properties in aqueous phase.

These experiments are crucial to differentiate this work from previous room temperature SF reports and to establish its practical significance in the field. Additionally, given the small size of these nanoparticles, there's potential for easy modification and use in solution phase, which could be explored further. Without these more applied demonstrations, the practical superiority of these nanoparticles for real-world imaging applications remains somewhat theoretical, and the full impact of the shortened lifetime and small particle size on actual biomedical imaging applications cannot be fully assessed.

Response:

We appreciate the reviewer for his/her suggestion to supplement the cell imaging experiment. The cell imaging experiment may be helpful, but it is beyond the scope of our current work, as our work focuses mainly on the photophysics of upconversion (UC) superfluorescence (SF) of lanthanide-doped UC nanoparticles (UCNPs). The main findings of our work regarding ultrafast UC-SF with a large number ($N = 912$) of correlated dipoles and an ultrashort lifetime of sub-2.5 ns are novel enough and had not been achieved before.

The imaging capability of UC-SF on the standard multiphoton microscopy system and its advantages over traditional UC luminescence (UCL) have been provided in Supplementary Fig. 17. The results showed that UC-SF can be coupled well with the standard multiphoton microscopy system and is bright enough for high-quality imaging, especially for fast scanning imaging. When the pixel dwell time was set as 1.1 μs , the images of UC microspheres (UCMSs) loaded with conventional $\text{NaYF}_4: \text{Yb/Er}@ \text{NaYF}_4$ (2Er@Y) UCNPs suffered from an obvious tailing effect due to the long radiative lifetime of normal UCL, which caused the image distortion and deformation. For comparison, the images of UCMSs loaded with $\text{NaNdF}_4@ \text{NaYF}_4$ (100Nd@Y) were clear and free from the tailing effect because of the ultrashort

Fujian Institute of Research on the Structure of Matter, Chinese Academy of Sciences

decay time of UC-SF. These results provide strong evidence that UC-SF with an ultrashort lifetime is superior to traditional UCL for high-speed scanning imaging, as also supported by Reviewer #2 in his/her comments. We sincerely hope the reviewer concurs after reading the above clarification.

Reply to the Comments of Reviewer #2

Comments:

The authors have effectively addressed the reviewers' comments, significantly enhancing the manuscript and making it suitable for publication. Additionally, I have been asked to comment on the application potential concerns raised by reviewer #1 and I think the experiments detailed in Supplementary Figure 17 adequately address these issues.

The imaging capabilities of upconversion superfluorescence (UC-SF) on a standard multiphoton microscopy system and its advantages over traditional UC luminescence are demonstrated in Supplementary Figure 17 of the revised Supplementary Information. The results show that UC microspheres, composed of NaNdF₄@NaYF₄ core-shell UCNPs coated on polystyrene, can be integrated with conventional multiphoton microscopy and provide sufficient brightness for high-quality imaging, especially for fast scanning applications. Moreover, the imaging performance of UC-SF is shown to be superior to traditional UC luminescence.

Response:

We greatly appreciate the reviewer for his/her positive comments on our manuscript, noting that it is suitable for publication.